# Plan for Speed: Dilated Scheduling for Masked Diffusion Language Models

**Omer Luxembourg** [1]   **Haim Permuter** [1]   **Eliya Nachmani** [1]

## Abstract

Masked diffusion language models (MDLMs) promise fast, non-autoregressive text generation, yet existing samplers, which pick tokens to unmask based on model confidence, ignore interactions when unmasking multiple positions in parallel and effectively reduce to slow, autoregressive behavior. We propose the Dilated Unmasking Scheduler (DUS), an inference-only, planner-model-free method that partitions sequence positions into non-adjacent dilated groups and unmasks them in parallel so as to minimize an upper bound on joint entropy gain at each denoising step. By explicitly trading off the number of network calls against generation quality, DUS recovers most of the performance lost under traditional parallel unmasking strategies. Across math (GSM8K, MATH500), code (HumanEval, MBPP), general-knowledge (BBH, MMLU-Pro), and instruction following (IFEval) benchmarks, DUS outperforms confidence-based planners and turns the diffusion-specific quality-speed trade-off into a deterministic, predictable speedup set by the block size $B$, yielding up to $5.8\times$ wall-clock speedup over token-by-token MDLM decoding without modifying the underlying denoiser. Applied as a drop-in post-filter, dilated spacing also improves adaptive samplers. Code is available at https://github.com/omerlux/DUS.

## 1. Introduction

Diffusion-based language models have emerged as a promising alternative to traditional autoregressive (AR) large language models (Grattafiori et al., 2024; Yang et al., 2025),

[1] School of Electrical and Computer Engineering, and Data Science Research Center, Ben-Gurion University of the Negev, Beersheba, Israel . Correspondence to: Omer Luxembourg <omerlux@post.bgu.ac.il>, Haim Permuter <haimp@bgu.ac.il>, Eliya Nachmani <eliyanac@bgu.ac.il>.

*Proceedings of the 43rd International Conference on Machine Learning*, Seoul, South Korea. PMLR 306, 2026. Copyright 2026 by the author(s).

offering potential advantages in parallel generation and controllability (Campbell et al., 2022; Lou et al., 2024; Sahoo et al., 2024). In discrete diffusion for text, generation is an iterative denoising process: starting from a fully noised (e.g., masked) sequence, the model reconstructs text over multiple steps. While this enables any-order generation, high quality often requires roughly one denoising pass per token, so generating length $G$ typically entails $\mathcal{O}(G)$ denoiser calls. Although current diffusion LMs still lag AR models on some benchmarks, this gap stems primarily from training maturity rather than fundamental architectural limitations: diffusion models outperform AR under data-scarce conditions (Prabhudesai et al., 2026), and a code-specialized diffusion LM has been reported to exceed its AR counterpart on HumanEval, MBPP, and BigCodeBench-Hard (Fan et al., 2026).

AR LLMs typically generate token-by-token, leading to linear inference latency. Speculative decoding partially mitigates this by using a lightweight draft model to propose multiple tokens that the full AR model verifies (Leviathan et al., 2023; Xia et al., 2022). By contrast, diffusion-based LMs reconstruct the entire sequence simultaneously: although naive masked denoising still incurs $\mathcal{O}(G)$ passes, it supports any-order and parallel generation, enabling inference schedules with fewer denoiser calls.

Existing strategies to accelerate diffusion sampling either introduce heuristic planners that select tokens to unmask based on confidence or entropy, or employ semi-AR block-wise generation that divides the sequence into contiguous spans and applies parallel diffusion within each block to preserve global coherence (Ye et al., 2025; Arriola et al., 2025; Nie et al., 2025b). However, planners can be shortsighted and prone to error propagation, and semi-AR diffusion still incurs $\mathcal{O}(B)$ denoiser calls per block of size $B$. Recent work has framed unmasking as an inference-time planning problem (Peng et al., 2025), incorporated planning mechanisms into the ELBO training objective (Liu et al., 2025), and improved efficiency via non-uniform schedules (Park et al., 2025) or denoiser-signal-based unmasking schemes (Ben-Hamu et al., 2025; Wu et al., 2025) that reveal multiple tokens in parallel.

In this work, we introduce the *Dilated Unmasking Scheduler* (DUS), a purely inference-time, model-agnostic planner that

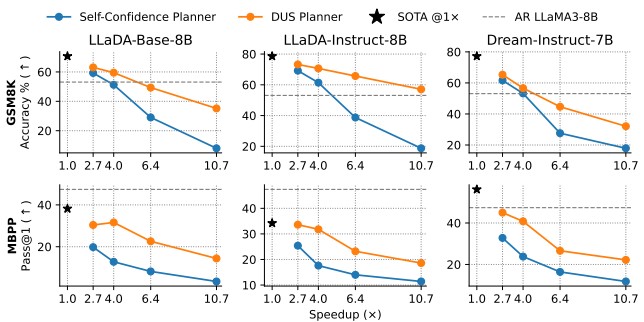

*(a)* Score vs. speedup - DUS and self-confidence MDLM planners.

*Figure 1.* (a) Trade-off between inference speedup and task accuracy on math (GSM8K, MATH500) and code (HumanEval, MBPP) benchmarks. Solid orange curves show DUS planner results; solid blue curves show self-confidence planner results. The star (⋆) marks the single-token, token-by-token SOTA baseline, and dashed gray lines denote AR LLaMA3-8B performance under the same protocol (Nie et al., 2025b; Ye et al., 2025; Grattafiori et al., 2024). (b), (c) Chain-of-thought on a GSM8K example with block size $B = 32$ (speedup 6.4×). Token shading (blue→orange) encodes the unmasking iteration. (b), DUS, generates a full, coherent reasoning trace; (c), self-confidence, truncates its chain-of-thought prematurely.

partitions each block of length $B$ into logarithmically many iterations using a fixed dilation pattern. Under a Markov assumption, this schedule minimizes the joint conditional entropy at each step, reducing denoiser calls from $\mathcal{O}(B)$ to $\mathcal{O}(\log B)$ without retraining or additional planner modules. Our main contributions are:

- **Inference-Only, Model-Agnostic Decoding:** A drop-in strategy requiring zero modifications to model architecture or training.

- **Logarithmic Unmasking Schedule:** Deterministic, coarse-to-fine dilation that respects local context and minimizes joint entropy, in $\mathcal{O}(\log B)$ denoiser iterations.

- **Theoretical Guarantees:** We show that DUS approaches the joint entropy bound at each iteration under fast-mixing assumptions, compared to baselines with the same step budget.

- **Empirical Validation:** Extensive experiments on math (GSM8K (Cobbe et al., 2021), MATH500 (Lightman et al., 2023)), code (HumanEval (Chen et al., 2021), MBPP (Austin et al., 2021)), general-knowledge (BBH (Suzgun et al., 2023), MMLU-Pro (Wang et al., 2024)), and instruction following (IFEval (Zhou et al., 2023)) benchmarks with LLaDA-8B, Dream-7B, DiffuCoder-7B demonstrate up to an order-of-magnitude reduction in denoiser calls and consistent quality improvements over confidence-based planners, along with a head-to-head comparison against adaptive entropy and confidence bounded samplers (Appendix B.9).

- **Complementary Hybrid Extension:** A drop-in dilated-spacing post-filter applied on top of adaptive samplers (EB/CB) improves their accuracy at matched

or modestly higher NFE, without modifying their score function or selection criterion (Section 4.4).

## 2. Related Work

Discrete diffusion sampling relies on a planner to decide which masked tokens to reveal at each reverse step (Liu et al., 2025). Early "self-planners" use the diffusion denoiser itself to rank positions by simple criteria: top-$k$ highest probability (confidence), lowest conditional entropy, or top-$k$ probability margin (the gap between the highest and second-highest scores) (Campbell et al., 2022; Sahoo et al., 2024; Kim et al., 2025). Recent analyses show that such confidence-based decoding empirically exhibits near-autoregressive, left-to-right ordering (Gong et al., 2025; Horvitz et al., 2025), limiting the parallelism advantage of diffusion models.

Path Planning (P2) extends beyond these self-planners by incorporating an external guiding model - a pretrained BERT - to evaluate candidate token sets according to their output probabilities (Peng et al., 2025). P2 compares denoiser-guided selection, random sampling, and BERT-scored planners, finding that the learned BERT planner yields better scores in various experiments at the cost of auxiliary model calls.

Semi-AR block diffusion segments a long sequence into contiguous spans of length $B$, runs denoiser iterations in parallel within each block, and reveals blocks sequentially (Arriola et al., 2025; Nie et al., 2025b). This approach does not reduce the total number of denoiser passes from $\mathcal{O}(G)$ to $\mathcal{O}(G/B)$ (where $G$ is the generation length), it still incurs $\mathcal{O}(B)$ calls per block, and it typically requires a policy to decide block order. While (Arriola et al., 2025) discusses both training and inference with semi-AR block diffusion,

here we focus exclusively on inference.

At the 7-8B parameter scale, LLaDA-8B (Nie et al., 2025b) (trained on 2.3T tokens) and Dream-7B (Ye et al., 2025) (with AR initialization) demonstrate the practical viability of fully parallel decoding, matching AR baselines on math, code, and reasoning benchmarks while offering supervised fine-tuned (SFT) variants for instruction following. DiffuCoder (Gong et al., 2025), trained solely on 400B code tokens, reduces left-to-right bias via temperature-controlled "causalness" and coupled-GRPO, enabling fully parallel decoding without semi-AR blocks.

Inference efficiency can also be improved orthogonally via Key-Value (KV) caching, which reuses stable transformer key/value tensors across denoising steps. Methods such as *FreeCache* (Hu et al., 2025), *Fast-dLLM* (Wu et al., 2025), and *dKV-Cache* (Ma et al., 2026) report up to $3\times$ speedups with minimal quality loss, at the cost of extra memory (and, for FreeCache, an auxiliary AR guide model).

Adaptive scheduling methods decide which tokens to unmask and when, aiming to improve inference speed while preserving quality. On the timestep axis, *Jump Your Steps* (JYS) selects a non-uniform subset of noise levels by estimating KL divergences between adjacent diffusion kernels, trading a small ELBO loss for fewer denoiser calls (Park et al., 2025). However, JYS targets score-matching diffusion models (e.g., SEDD (Lou et al., 2024)), which currently operate at smaller scale, precluding meaningful comparison on our benchmarks. Moreover, many discrete masked diffusion models are trained with cross-entropy objectives (rather than continuous score-based losses) (Sahoo et al., 2024), making direct log-likelihood scheduling less straightforward. At the token level, entropy-bounded (EB) sampling selects the largest set of tokens whose cumulative conditional entropy is below a user-specified bound, achieving $2\text{-}3\times$ speedups without retraining (Ben-Hamu et al., 2025); confidence-bounded (CB) variants show similar gains (Wu et al., 2025). These per-token heuristics are effective but do not explicitly model token dependencies when unmasking in parallel. We revisit EB and CB both as baselines (Appendix B.9) and as a substrate on which the dilated-spacing principle acts as a drop-in post-filter (Section 4.4).

## 3. Method

We cast masked diffusion inference as the interaction of a fixed denoiser $\mathcal{D}_\theta$ and a planner $\mathcal{P}$ that selects which masked positions to reveal at each iteration. Building on this view, we propose *Dilated Unmasking Scheduling* (DUS), a deterministic coarse-to-fine planner for block-wise (semi-autoregressive) masked diffusion that reduces the number of denoiser evaluations per block from $\mathcal{O}(B)$ to $\mathcal{O}(\log B)$ while preserving generation quality.

We begin with notation and the denoiser-planner decomposi-

tion, derive an entropy-based surrogate objective for planner design in the block-wise setting, review denoiser-guided self-planners, present DUS, and finally give theoretical guarantees (with proofs deferred to the appendix).

### 3.1. Notation

Let $\mathcal{X} = \{X_1, \ldots, X_G\}$ be a length-$G$ sequence of random variables, each taking values in a finite token vocabulary $\mathcal{V}$ of size $|\mathcal{V}|$. At each decoding iteration step $t$, we represent the partially observed (masked) sequence by $\mathcal{M}_t = (M_{t,1}, \ldots, M_{t,G})$, where $M_{t,i} = X_i$ if position $i$ is currently unmasked and $M_{t,i} = \texttt{[MASK]}$ otherwise. We write $\mathcal{S}_t$ for the sampler state at iteration $t$ (equivalently, the information contained in $\mathcal{M}_t$).

For our theoretical analysis we will assume that $\mathcal{X}$ is generated by a stationary, ergodic variable-length Markov chain (VLMC) with fast mixing; the formal assumption is stated in Appendix C.3.

**Denoiser and planner.** A masked diffusion sampler can be viewed as the interaction of: (i) a *denoiser* $\mathcal{D}_\theta$ (with parameters $\theta$) that, given the current partially masked state $\mathcal{S}_t$, outputs token-wise categorical distributions $p_\theta(X_i \mid \mathcal{S}_t) \in \Delta(\mathcal{V})$ for each currently masked position, where $\Delta(\mathcal{V})$ denotes the probability simplex over $\mathcal{V}$, and (ii) a *planner* $\mathcal{P}_t \colon \mathcal{S}_t \times \mathcal{D}_\theta \to 2^{\{1,\ldots,G\}}$ that chooses which masked indices to reveal next. Importantly, the planner may use *denoiser-derived signals* (e.g., per-token probabilities, entropies, or other confidence scores), which we write as

$$\mathcal{I}_t = \mathcal{P}_t(\mathcal{S}_t; \mathcal{D}_\theta) \subseteq \{1, \ldots, G\}. \tag{1}$$

The planner controls only *which indices* are revealed. The *values* revealed at those indices are then sampled from the denoiser distributions, after which the state is updated to obtain $\mathcal{S}_{t+1}$.

Masked diffusion language models are trained by maximizing a masked cross-entropy ELBO that upper-bounds $(-\log p_\theta(x))$ (Sahoo et al., 2024; Shi et al., 2024; Campbell et al., 2022; Nie et al., 2025b); the formal forward process and training objective are given in Appendix A.2. At inference time, the denoiser parameters $\theta$ are fixed, and the only remaining degree of freedom is the planner (i.e., the sequence of index sets $\{\mathcal{I}_t\}$). We formalize the planner objective in the block-wise setting in Section 3.3.

### 3.2. MDLM Inference in Practice

In large-scale discrete diffusion language models, inference is often implemented in a semi-autoregressive (semi-AR), block-wise manner: the sequence is partitioned into consecutive blocks of size $B$, and each block is completed via several denoising rounds before moving to the next (Nie

et al., 2025b). Block diffusion variants typically train the denoiser by masking contiguous spans of length $B$ and recovering them jointly, which improves local coherence, while other variants train on masks spread across the entire sequence (Arriola et al., 2025). We next formalize an entropy-based surrogate objective for choosing $\mathcal{I}_t$ in this block-wise setting.

### 3.3. Detailed Formulation

While the denoiser is trained over uniformly sampled masking rates (Sahoo et al., 2024; Shi et al., 2024), the inference-time masking pattern, a fully observed prompt followed by a contiguous block of masked tokens, is an edge case of the training distribution. The denoiser is therefore not well-calibrated for this regime (Kim et al., 2025), and multi-step unmasking improves sample quality. Consequently, inference unfolds over multiple unmasking-denoising iterations (Campbell et al., 2022; Nie et al., 2025a). Large-scale diffusion LLMs typically partition the sequence into semi-AR blocks of length $B$ and apply several denoiser iterations within each block before moving to the next (Nie et al., 2025b).

**Block-wise setting.** At iteration $t$, let $\mathcal{S}_t$ denote the full state of the sequence: (1) all previously unmasked blocks, (2) the current block containing a mixture of masked and unmasked tokens, and (3) the remaining future blocks that are still fully masked. Denote the indices of the current block by $\{b, \dots, b + B - 1\}$. The planner selects a set of indices

$$\mathcal{I}_t = \mathcal{P}_t(\mathcal{S}_t; \mathcal{D}_\theta) \subseteq \{b, \dots, b + B - 1\}, \quad (2)$$

where the dependence on $\mathcal{D}_\theta$ captures that the planner may use denoiser-derived signals (e.g., confidence/entropy). After sampling $\hat{X}_{\mathcal{I}_t}$ from the denoiser distributions, the state is updated to $\mathcal{S}_{t+1}$ and the denoiser runs again, yielding new conditionals due to the updated context.

**Optimal estimator and parallel sampling.** We analyze planner design under an *optimal estimator* assumption, used purely as an analysis device: the denoiser model's distribution $p_\theta$ at each iteration $t$ outputs the data conditional distribution for every currently masked position.

**Assumption 3.1** (Optimal estimator). For every iteration $t$ and every masked index $i$, the denoiser provides $p_\theta(X_i \mid \mathcal{S}_t)$ and equals the real data distribution $p(X_i \mid \mathcal{S}_t)$.

From this point on, under Assumption 3.1, we write $p$ as the optimal estimator distribution and drop the subscript $\theta$ from entropies and mutual information (MI) (i.e., we write $H$ and $I$). In practice, a learned denoiser only approximates these conditionals. Our goal here is to isolate the role of the planner given $p$.

**Assumption 3.2** (Planner objective). Fix an iteration $t$ and a state $\mathcal{S}_t$. For any candidate set $\mathcal{I}_t = \{i_1^{(t)}, \dots, i_{k_t}^{(t)}\}$, the

planner's ideal objective is the expected joint negative log-likelihood

$$\mathcal{J}(\mathcal{I}_t; \mathcal{S}_t) := \mathbb{E}_{X_{\mathcal{I}_t} \sim p(\cdot \mid \mathcal{S}_t)}\big[- \log p(X_{\mathcal{I}_t} \mid \mathcal{S}_t)\big] \quad (3)$$

We write this objective as an expectation since under stationarity and ergodicity the long-run average log loss equals its expectation.

Note that this expected joint log loss is equal to the conditional joint entropy

$$\mathbb{E}_{X_{\mathcal{I}_t} \sim p(\cdot \mid \mathcal{S}_t)}\big[- \log p(X_{\mathcal{I}_t} \mid \mathcal{S}_t)\big] = H(X_{\mathcal{I}_t} \mid \mathcal{S}_t)$$
$$= H\big(X_{i_1^{(t)}}, \dots, X_{i_{k_t}^{(t)}} \mid \mathcal{S}_t\big). \quad (4)$$

**Parallel sampling.** Given a selected set $\mathcal{I}_t = \{i_1^{(t)}, \dots, i_{k_t}^{(t)}\}$, a *parallel sampler* reveals all tokens in $\mathcal{I}_t$ simultaneously by sampling them independently from the denoiser conditionals. Equivalently, it uses the product-of-marginals approximation $p^{\mathrm{par}}(X_{\mathcal{I}_t} \mid \mathcal{S}_t) := \prod_{j=1}^{k_t} p(X_{i_j^{(t)}} \mid \mathcal{S}_t)$. Under this sampler, the step-$t$ objective induced by parallel sampling is the expected negative log-likelihood

$$\mathcal{L}(\mathcal{I}_t; \mathcal{S}_t) := \mathbb{E}_{X_{\mathcal{I}_t} \sim p^{\mathrm{par}}(\cdot \mid \mathcal{S}_t)}\big[- \log p^{\mathrm{par}}(X_{\mathcal{I}_t} \mid \mathcal{S}_t)\big]$$
$$= \sum_{j=1}^{k} H(X_{i_j^{(t)}} \mid \mathcal{S}_t). \quad (5)$$

This sum-of-marginals objective, $\mathcal{L}(\mathcal{I}_t; \mathcal{S}_t)$, is the tractable surrogate optimized by planners. We next relate the tractable surrogate to the intractable joint objective $H(X_{\mathcal{I}_t} \mid \mathcal{S}_t)$.

**Lemma 3.3** (Token-by-token vs. parallel bounds). *Let the current block indices be $\mathcal{B} := \{b, \dots, b + B - 1\}$. Consider any partition of the block into $T \in \mathbb{N}$ disjoint groups $\mathcal{I}_1, \dots, \mathcal{I}_T$ with $\bigcup_{t=1}^{T} \mathcal{I}_t = \mathcal{B}$, and define $\mathcal{I}_{<t} := \bigcup_{s<t} \mathcal{I}_s$. Under the optimal estimator assumption, the total loss induced by parallel sampling within each group is $\sum_{t=1}^{T} \mathcal{L}\big(\mathcal{I}_t; \mathcal{S}_1, X_{\mathcal{I}_{<t}}\big)$, then*

$$H(X_{\mathcal{B}} \mid \mathcal{S}_1) \leq \sum_{t=1}^{T} \mathcal{L}\big(\mathcal{I}_t; \mathcal{S}_1, X_{\mathcal{I}_{<t}}\big) \leq \sum_{i \in \mathcal{B}} H(X_i \mid \mathcal{S}_1).$$
$$(6)$$

*The left inequality is tight and is achieved by token-by-token unmasking (all groups have size $|\mathcal{I}_t| = 1$), while the right inequality corresponds to full parallel sampling (a single group $\mathcal{I}_1 = \mathcal{B}$). A proof is given in Appendix C.1.*

**Corollary 3.4** (Gap-minimization principle for parallel sampling). *For any iteration $t$ and any parallel (grouped) sampling step that reveals a set $\mathcal{I}_t$ simultaneously, define the entropy gap*

$$\Delta(\mathcal{I}_t; \mathcal{S}_t) := \sum_{i \in \mathcal{I}_t} H(X_i \mid \mathcal{S}_t) - H(X_{\mathcal{I}_t} \mid \mathcal{S}_t), \quad (7)$$

and $\Delta(\mathcal{I}_t; \mathcal{S}_t) \geq 0$ *(by Lemma 3.3). Thus, to make parallel sampling behave like the joint objective, planner design should explicitly minimize the entropy gap $\Delta(\mathcal{I}_t; \mathcal{S}_t)$.*

Corollary 3.4 suggests the key design goal for any parallel sampler: choose $\mathcal{I}_t$ to minimize the entropy gap $\Delta(\mathcal{I}_t; \mathcal{S}_t)$.

**Design principle.** This motivates planners that avoid selecting strongly coupled (typically nearby) indices in the same iteration conditioned on $S_t$. DUS enforces spacing to reduce conditional dependence, thereby reducing $\Delta(\mathcal{I}_t; \mathcal{S}_t)$ and making the marginal-based surrogate a faithful proxy for the joint.

We now describe denoiser-guided self-planners and then present DUS as a deterministic alternative, with planner design guided by keeping the surrogate in (5) close to the joint objective by controlling the entropy gap $\Delta(\mathcal{I}_t; \mathcal{S}_t)$.

### 3.4. Self-Planners Guided by Denoiser Confidence

Self-planners are denoiser-guided planners that select indices to reveal using per-token scores derived from the denoiser conditionals. At iteration $t$, each masked index $i$ is assigned a score $r(i \mid \mathcal{S}_t)$ computed from $p(X_i \mid \mathcal{S}_t)$, and the planner selects an index set $\mathcal{I}_t$ (e.g., top-$k$ or a threshold rule) (Kim et al., 2025; Wu et al., 2025; Ben-Hamu et al., 2025). Two common instantiations are: (i) *confidence-based* scoring, $r(i \mid \mathcal{S}_t) = \max_x p(x_i \mid \mathcal{S}_t)$, and (ii) *entropy-based* scoring, $r(i \mid \mathcal{S}_t) = H(x_i \mid \mathcal{S}_t)$.

Because these scores are per-token, self-planners effectively minimize the parallel-sampling surrogate in (5). However, they do not explicitly control the dependence among indices selected in the same iteration, and thus may incur a large entropy gap $\Delta(\mathcal{I}_t; \mathcal{S}_t)$ when $\mathcal{I}_t$ contains strongly coupled (often nearby) positions.

### 3.5. DUS as Predefined Planner

DUS is a deterministic, inference-only planner that fixes in advance which positions will be unmasked at each iteration, independently of model outputs. In the core, DUS directly specifies the index set to reveal at each iteration, denoted $\mathcal{I}_t$, and reveals all indices in $\mathcal{I}_t$ in parallel.

Given a block of size $B$ and a base $a > 1$, DUS completes the block in $R = \lceil \log_a B \rceil$ denoiser evaluations (versus $\mathcal{O}(B)$ for token-by-token unmasking).

**Implementation details.** In practice, DUS can be augmented with a *skip* heuristic that defers unmasking positions in $\mathcal{I}_t$ whose denoiser scores indicate high uncertainty. Unless stated otherwise, we use the core schedule in (8)-(9) without skip. An empirical sweep over the skip threshold is in Appendix B.2.

Inside a block, positions are indexed by $\{1, \ldots, B\}$, and the number of iterations and step sizes are defined as

$$R = \lceil \log_a B \rceil, \quad s_t = \lfloor \tfrac{B}{a^t} \rfloor, \quad t = 1, \ldots, R. \quad (8)$$

The scheduled $\mathcal{I}_t$ at iteration $t$ and the incremental unmasked set $\mathcal{U}_t$ are defined as

$$\mathcal{I}_t = \big\{ k \in \{1, \ldots, B\} \setminus \mathcal{U}_{t-1} \mid (k-1) \bmod s_t = 0 \big\},$$
$$\mathcal{U}_0 = \emptyset, \quad \mathcal{U}_t = \mathcal{U}_{t-1} \cup \mathcal{I}_t. \quad (9)$$

If $|\mathcal{I}_R| < |\mathcal{I}_{R-1}|$, then $\mathcal{I}_R$ is merged into $\mathcal{I}_{R-1}$ to balance coverage. This schedule completes a block in $R \approx \lceil \log_a B \rceil$ iterations, where the values $s_t$ make early iterations unmask a few widely spaced positions and later iterations fill in the remaining gaps at finer resolution. For example, let $B = 8$ and $a = 2$. Then $R = 3$, $s_1 = 4$, $s_2 = 2$, $s_3 = 1$, and

$$\mathcal{I}_1 = \{ k \mid (k-1) \bmod 4 = 0 \} = \{1, 5\},$$
$$\mathcal{I}_2 = \{ k \notin \{1, 5\} \mid (k-1) \bmod 2 = 0 \} = \{3, 7\},$$
$$\mathcal{I}_3 = \{ k \notin \{1, 3, 5, 7\} \mid (k-1) \bmod 1 = 0 \} = \{2, 4, 6, 8\}.$$

### 3.6. Theoretical Analysis of DUS

Corollary 3.4 shows that parallel unmasking is accurate when the simultaneously revealed indices have a small entropy gap $\Delta(\mathcal{I}_t; \mathcal{S}_t)$, i.e., when the tractable surrogate $\sum_{i \in \mathcal{I}_t} H(X_i \mid \mathcal{S}_t)$ is close to the intractable joint entropy $H(X_{\mathcal{I}_t} \mid \mathcal{S}_t)$. Under a fast-mixing VLMC model, spacing the indices in $\mathcal{I}_t$ makes their conditional dependence small, which in turn makes this gap small. The next lemma formalizes that, for sufficiently well-separated indices, the joint conditional entropy is $\varepsilon$-close to the sum of individual conditional entropies.

**Lemma 3.5.** *Assume that $(X_1, \ldots, X_G)$ is a stationary, ergodic VLMC with finite order $L$ with fast-mixing property. Let $\mathcal{I}_t = \{i_1^{(t)}, \ldots, i_{k_t}^{(t)}\} \subseteq \{1, \ldots, B\}$ be indices such that the pairwise distances satisfy $\big|i_m^{(t)} - i_n^{(t)}\big| \geq d$ for all $m \neq n$. Then, for any $\varepsilon > 0$, there exists a distance threshold $D_\varepsilon$ such that for all $d \geq D_\varepsilon$,*

$$H\big(X_{i_1^{(t)}}, \ldots, X_{i_{k_t}^{(t)}} \mid \mathcal{S}_t\big) \geq \sum_{j=1}^{k_t} H\big(X_{i_j^{(t)}} \mid \mathcal{S}_t\big) - \varepsilon. \quad (10)$$

The proof of Lemma 3.5 relies on the fast-mixing property of the underlying VLMC, which implies that the MI between a token and any finite block of sufficiently distant tokens decays rapidly with their separation. The following lemma formalizes this decay. Restated lemma and proof are given in Appendix C.2.

**Lemma 3.6** (MI decays under fast mixing). *Let $(X_t)_{t \in \mathbb{Z}}$ be a stationary, ergodic VLMC with finite order $L < \infty$, fast-mixing property on a finite alphabet (Assumption C.1*

*in Appendix C.3). Then there exist constants $C < \infty$ and $\rho \in (0, 1)$ such that for any index $i$, spacing $d \geq 1$, and any finite $M \geq 0$,*

$$I\big(X_i; X_{i+d}, X_{i+2d}, \ldots, X_{i+(M+1)d}\big) \;\leq\; C\,\rho^{\,d}. \quad (11)$$

*A proof of* (11) *is given in Appendix C.3.*

In the context of DUS, the indices in $\mathcal{I}_t$ within a fixed iteration $t$ satisfy a minimum separation $s_t$ in index space (Section 3.5). Combined with Lemma 3.6, this spacing implies that the MI between selected positions can be made uniformly small when $s_t$ is large, which in turn yields the joint-entropy bound in Lemma 3.5 for the set $\mathcal{I}_t$.

**Late iterations and richer context.** The bound is most informative in the early, coarse iterations of DUS where the spacing $s_t$ is large. As $t$ increases, DUS necessarily decreases $s_t$ to fill finer-grained gaps. In this regime, nearby tokens can be strongly dependent and we do not claim conditional independence. However, by late iterations the state $\mathcal{S}_t$ contains substantially more revealed neighbors around each remaining mask, providing richer surrounding context to the denoiser. This typically lowers the remaining per-token conditional entropies and helps the model correct local inconsistencies introduced by earlier parallel decisions.

In summary, DUS aims to make the marginal-entropy surrogate in (5) a faithful proxy for the joint objective by explicitly controlling the entropy gap $\Delta(\mathcal{I}_t; \mathcal{S}_t)$ through spacing. In contrast, denoiser-guided self-planners based on per-token scores can select tightly clustered indices in the same iteration, for which the surrogate may be a loose upper bound on the joint uncertainty. Two features of the predefined DUS schedule are designed to reduce this mismatch:

1. **Maintained spacing across iterations.** Early iterations reveal a sparse set of indices with large separation, which (under fast mixing) reduces within-step dependence among simultaneously sampled tokens.

2. **Coarse-to-fine contextual conditioning.** By spreading early reveals across the block, later iterations predict remaining masks given richer local context from multiple regions of the block.

Under the fast-mixing VLMC assumption, this design makes the entropy gap small in the coarse iterations, and therefore supports the use of parallel unmasking within each $\mathcal{I}_t$.

When long-range dependencies are strong (e.g., global constraints in code or poetry), any parallel unmasking scheme can still sample dependent tokens together and incur a larger entropy gap. In such settings, DUS should be viewed as a heuristic that trades fewer denoiser evaluations for approximate parallel decisions, with later iterations potentially

correcting some local inconsistencies via additional context. The VLMC fast-mixing assumption serves primarily as an analytical device motivating the dilated design, not a literal claim about all data. DUS only requires tokens spaced at distance $d$ to be on average less dependent, a substantially milder block-level requirement than full-sequence fast mixing. Empirically, DUS attains 2-3$\times$ wider average token spacing than alternative planners while achieving equal or higher accuracy on HumanEval and MATH500 (Appendix B.7), validating the schedule on domains that violate strict fast mixing.

### 3.7. Experimental Setup

For each model and dataset, decoding uses a semi-AR masked diffusion process with block sizes $B \in \{8, 16, 32, 64\}$. Decoding proceeds in $n_{\text{blocks}} = G/B$ rounds; within each round, the model iteratively predicts the masked tokens of the current block before moving to the next. With DUS, the average number of function evaluations per block is $\text{NFE}_{\text{block}} = \log_2 B$, so the average parallelism is $k = B/\log_2 B$. The total number of denoiser calls (total NFE) is

$$\text{NFE} = n_{\text{blocks}}\,\text{NFE}_{\text{block}} = \frac{G}{B}\log_2 B, \quad (12)$$

so larger blocks trade more parallelism for fewer diffusion evaluations. Section 4.3 further studies fixed-$k$ versus incremental-$k$ (DUS-like) schedules for self-confidence and random planners; aside from that ablation, all non-DUS planners use fixed $k$.

We evaluate five MDLM variants: (1) LLaDA-Base-8B and (2) LLaDA-Instruct-8B (Nie et al., 2025b), an 8B mask-diffusion transformer pretrained on mixed text+code and its instruction-tuned version. (3) Dream-Instruct-7B (Ye et al., 2025), a 7B instruction-tuned model; we focus on this variant since Dream-Base performs substantially worse in our setup. (4) DiffuCoder-Base-7B and (5) DiffuCoder-Instruct-7B (Gong et al., 2025), a 7B model trained on code tokens with AR initialization and its instruction-tuned version. The DiffuCoder authors report results without a semi-AR protocol, thus, for DiffuCoder baselines (without DUS) we disable semi-AR inference. Four planners are evaluated:

1. **Self-confidence (baseline).** At each block the model's top-$k$ confident tokens are unmasked at each diffusion reverse iteration, while the others are masked. $k$ is set to a fixed value that is dependent on block size of the semi-AR process, $k = \log_2 B$, unless stated otherwise. LLaDA and Dream models use maximum probability as their confidence while DiffuCoder uses entropy (as in their original work).

*Table 1.* Math (GSM8K, MATH500) and code (Humaneval, MBPP) benchmarks for self-confidence (Conf.) and DUS (ours). Tasks reported accuracy (%) for math and pass@1 for code, at block sizes $B = \{8, 16, 32, 64\}$, with corresponding speedup factor ($\times$). Model names: B=Base, I=Instruct. DiffuCoder is tested only on code benchmarks: HumanEval and MBPP. **Bold** marks better planner scores. $*$ denotes our own reruns for token-by-token MDLM baselines, †Llama-3-8B, and ‡Qwen-3-8B.

| | Model | ×1 | B=8 ×2.7 | | B=16 ×4 | | B=32 ×6.4 | | B=64 ×10.7 | | AR Baseline |
| | | Conf. | Conf. | DUS | Conf. | DUS | Conf. | DUS | Conf. | DUS | L† / Q‡ |
| --- | --- | --- | --- | --- | --- | --- | --- | --- | --- | --- | --- |
| GSM8K | LLaDA-B | 72.63* | 59.29 | **63.08** | 51.23 | **59.51** | 29.04 | **49.36** | 8.04 | **35.18** | |
| | LLaDA-I | 80.29* | 69.22 | **73.24** | 61.41 | **70.66** | 38.74 | **65.73** | 18.73 | **57.09** | 49.81* / 88.55* |
| | Dream-I | 77.10* | 61.64 | **65.28** | 53.22 | **56.63** | 27.60 | **44.66** | 17.89 | **32.07** | |
| MATH500 | LLaDA-B | 24.00* | 16.6 | **21.4** | 11.2 | **19.2** | 6.0 | **13.6** | 2.6 | **10.2** | |
| | LLaDA-I | 28.80* | 21.4 | **23.8** | 15.4 | **22.8** | 10.8 | **19.2** | 8.0 | **14.8** | 15.23* / 50.20* |
| | Dream-I | 37.00* | 22.4 | **27.0** | 15.4 | **19.8** | 7.2 | **13.2** | 4.0 | **11.6** | |
| Humaneval | LLaDA-B | 34.76* | 15.85 | **25.61** | 12.8 | **19.51** | 4.88 | **14.02** | 4.88 | **6.71** | |
| | LLaDA-I | 39.02* | 21.95 | **28.05** | 14.02 | **23.17** | 9.76 | **10.37** | 10.98 | **11.59** | |
| | Dream-I | 57.90 | 8.54 | **14.63** | 5.49 | **11.59** | **6.71** | 6.71 | 6.10 | **9.15** | 36.59* / 61.59* |
| | DiffuCoder-B | 67.10 | 17.07 | **28.66** | 6.71 | **38.41** | 2.44 | **21.95** | 0.61 | **6.10** | |
| | DiffuCoder-I | 72.00 | 7.93 | **22.56** | 14.02 | **20.12** | **13.41** | 12.80 | **11.59** | 8.54 | |
| MBPP | LLaDA-B | 38.0* | 19.8 | **30.4** | 12.8 | **31.6** | 8.2 | **22.6** | 3.4 | **14.4** | |
| | LLaDA-I | 39.4* | 25.4 | **33.6** | 17.6 | **31.8** | 14.0 | **23.2** | 11.4 | **18.6** | |
| | Dream-I | 56.2 | 32.8 | **45.0** | 23.8 | **40.8** | 16.4 | **26.6** | 11.8 | **22.2** | 48.4* / 65.4* |
| | DiffuCoder-B | 74.2 | 29.2 | **48.6** | 17.4 | **43.0** | 10.2 | **27.4** | 3.4 | **17.2** | |
| | DiffuCoder-I | 75.1 | 31.8 | **46.4** | 25.6 | **43.6** | 21.0 | **26.6** | 13.0 | **18.2** | |

*Table 2.* General knowledge (BBH, MMLU-pro) benchmarks for self-confidence (Conf.) and DUS (ours). Both are few-shot, COT, multiple-choice datasets on general topics from various fields. Tasks report accuracy (%) for token-by-token ($\times 1$) and block sizes $B = \{8, 16, 32, 64\}$, with corresponding speedup factor ($\times$). Model names: B=Base, I=Instruct. For token-by-token baselines, $*$ denotes our own reruns. **Bold** marks better planner scores. †Llama-3-8B; ‡Qwen-3-8B.

| | Model | ×1 | B=8 ×2.7 | | B=16 ×4 | | B=32 ×6.4 | | B=64 ×10.7 | | AR Baseline |
| | | Conf. | Conf. | DUS | Conf. | DUS | Conf. | DUS | Conf. | DUS | L† / Q‡ |
| --- | --- | --- | --- | --- | --- | --- | --- | --- | --- | --- | --- |
| BBH | LLaDA-B | 44.26* | 41.67 | **43.33** | 40.93 | **41.48** | 35.93 | **40.56** | 26.67 | **38.52** | |
| | LLaDA-I | 53.89* | 51.48 | **51.67** | 47.59 | **50.37** | 44.26 | **50.93** | 41.67 | **50.37** | 52.59* / 59.63* |
| | Dream-I | 58.15* | 55.93 | **54.81** | 50.93 | **53.52** | 47.04 | **50.56** | 36.30 | **37.41** | |
| MMLU-Pro | LLaDA-B | 39.82* | 36.07 | **41.96** | 32.14 | **37.32** | 24.11 | **32.50** | 16.07 | **30.36** | |
| | LLaDA-I | 40.89* | 34.46 | **35.89** | 25.71 | **34.64** | 31.07 | **34.64** | 16.07 | **32.68** | 37.86* / 52.68* |
| | Dream-I | 50.89* | 46.43 | **48.04** | 36.96 | **47.32** | 27.50 | **48.39** | 13.75 | **39.64** | |

2. **EB- and CB-Sampler (literature baselines).** Adaptive samplers that unmask tokens by entropy bound $\gamma$ (Ben-Hamu et al., 2025) or confidence threshold $\tau$ (Wu et al., 2025); we sweep $\gamma$ and $\tau$ (Appendix B.9).

3. **DUS (ours).** For each block length $B$ the DUS is applied (as defined in Section 3.5), that unmasks on average $k = \log_2 B$ tokens in denoiser iteration, across a block.

4. **Adaptive samplers with dilated spacing (ours).** DUS spacing applied as a post-filter on EB or CB selections (Section 4.4).

Experiments report block size $B$ and relative NFE speedup, $B/\log_2 B$ (originates from (12)), defined as the ratio of token-by-token total NFE to the experiment's total NFE; planners are compared at matched $B$ and NFE budget, with their final task scores reported accordingly. A GSM8K problem, for $B = 32$, is visualized in Figure 1b and Figure 1c for DUS and self-confidence respectively.

## 4. Experiments

We evaluate the generative and downstream performance of DUS on mathematical reasoning, code generation, and general-knowledge chain-of-thought (COT) multiple-choice tasks, using unified benchmarking protocols for three diffusion LLMs (LLaDA, Dream, and DiffuCoder). Full dataset and evaluation details are provided in Appendix A.3. We first present math and coding results (Section 4.1), then general-knowledge reasoning (Section 4.2), and finally ablations on planner parallelism and NFE (Section 4.3).

### 4.1. Math and Coding Experiments

We evaluate GSM8K (Cobbe et al., 2021), MATH500 (Lightman et al., 2023), HumanEval (Chen et al., 2021), and MBPP (Austin et al., 2021) to study the speed-accuracy trade-off of semi-AR masked diffusion. Figure 1a plots accuracy vs. speedup for $2.7\times$, $4\times$, $6.4\times$, and $10.7\times$ (block sizes $B \in \{8, 16, 32, 64\}$): smaller blocks yield higher accu-

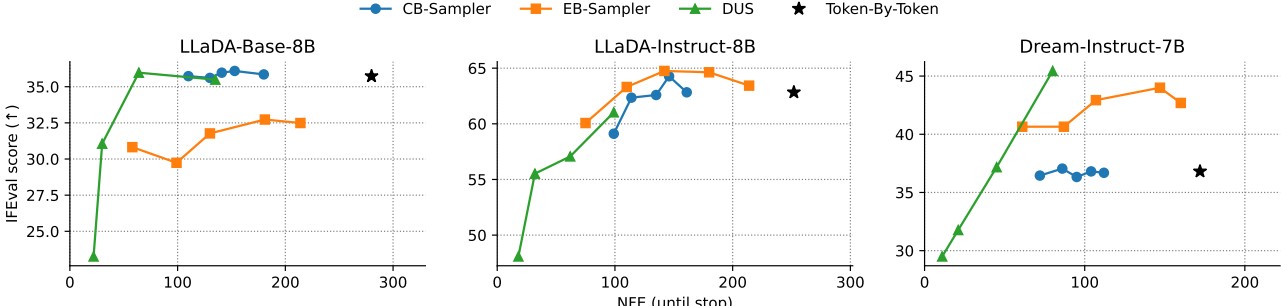

*Figure 2.* IFEval score versus NFE (for early termination) for CB-Sampler (blue), EB-Sampler (orange), and DUS (green) on LLaDA-Base-8B, LLaDA-Instruct-8B, and Dream-Instruct-7B. CB varies the confidence temperature $\tau = \{0.5, 0.6, 0.7, 0.9\}$, EB varies the entropy bound $\gamma = \{0.01, 0.1, 1, 2, 4\}$, and DUS varies the block size $B = \{8, 16, 32, 64\}$. For comparison, AR model, Llama-3-8B achieves score of 15.23 with 356 NFEs on average, and Qwen-3-8B achieves 41.73 with 577 NFEs on average. $\star$ denotes token-by-token greedy sampler. Higher accuracy (%) and lower NFE are better.

racy but require more denoiser rounds (higher NFE), while larger blocks enable up to $10\times$ fewer iterations with some loss in final accuracy. At the same NFE budget, DUS improves performance by up to 27% over the self-confidence baseline. GSM8K degrades smoothly with speed, whereas MBPP trends are less smooth (likely due to already low accuracy at higher speedups), but DUS remains consistently better.

Table 1 reports results across all four benchmarks. Overall, DUS matches or improves the self-confidence baseline, reaching up to 40% gains on math and up to 20% on code, while using less NFE than token-by-token inference. For reference, we also report token-by-token self-confidence baselines ($\times 1$ speedup).

Notably, on code benchmarks, the LLaDA and DiffuCoder base models can outperform their instruct counterparts under DUS, in contrast to the self-confidence planner where instruct tuning is generally superior.

### 4.2. General Knowledge Experiments

We evaluate general-knowledge reasoning on BBH (Suzgun et al., 2023) and MMLU-Pro (Wang et al., 2024) using a few-shot chain-of-thought protocol. Table 2 reports token-by-token self-confidence baselines ($\times 1$) and semi-AR block diffusion results for $B \in \{8, 16, 32, 64\}$ on LLaDA-Base, LLaDA-Instruct, and Dream-Instruct. Although gains are smaller than for math and code, DUS consistently outperforms self-confidence at matched speedups (BBH: +1-5%; MMLU-Pro: +5-9%), confirming stronger reasoning under constrained inference budgets.

### 4.3. Planners Parallelism Performance

**Power of spreading.** To isolate whether DUS helps due to its *schedule* (rather than simply revealing more tokens per step), we compare two schedules with the same average par-

allelism $k = \log_2 B$: (i) *fixed-k* (constant $k$ each step) and (ii) *dilated-incremental* (small $k$ early, larger $k$ late; DUS-like), conceptually related to cosine schedules (Shi et al., 2024). We evaluate GSM8K accuracy (first 300 samples) for LLaDA-Base and LLaDA-Instruct at $B \in \{16, 32\}$ in Table 3.

DUS performs best overall. For self-confidence, fixed-$k$ slightly outperforms dilated-incremental, indicating that confidence-guided selection benefits from uniform parallelism. For random planners, dilated-incremental recovers much of the fixed-$k$ accuracy loss and can even surpass fixed-$k$ self-confidence. Overall, spacing-aware scheduling (not just larger $k$) is key for maintaining accuracy under aggressive parallel decoding.

**Guaranteed Speedups.** We evaluate instruction following on IFEval (Zhou et al., 2023). Figure 2 plots score vs. NFE until early stop (first `<EOS>`) for CB-Sampler, EB-Sampler, and DUS on LLaDA-Base-8B, LLaDA-Instruct-8B, and Dream-Instruct-7B (generation length 1024). CB/EB vary confidence/entropy thresholds, while DUS varies block size $B$.

Across the three models, DUS reaches near-token-by-token accuracy at $3\text{-}4\times$ fewer NFE, with EB and CB occupying higher-NFE operating points (Figure 2).

Across a broader 5-model $\times$ 4-benchmark sweep (Appendix B.9), the adaptive samplers reach higher accuracy at higher NFE on most tasks, while DUS provides a predictable, deterministic speedup - often beating the nominal $(G/B) \log B$ budget - motivating the question of whether DUS-style spacing can be combined with adaptive selection (Section 4.4).

**Practical Considerations.** DUS is most effective with moderate block sizes ($B = 16\text{-}32$), where the dilated pat-

*Table 3.* Planners parallelism ablation on GSM8K (300 samples, $G = 256$), comparing self-confidence (Conf.) vs. random planners under two unmasking schedules: fixed-$k$ and dilated-incremental (Inc., as in DUS). Accuracy (%) is shown for both base and instruct LLaDA models at $B = \{16, 32\}$. Best scores are **bold**, second-best are underlined.

| Model | Block | Conf. | | Random | | DUS |
|---|---|---|---|---|---|---|
| | | fixed | inc. | fixed | inc. | inc. |
| Base | 16 | 55.00 | 49.33 | 24.67 | 43.67 | **61.33** |
| | 32 | 30.67 | 25.00 | 10.00 | 39.00 | **48.33** |
| Instruct | 16 | 62.33 | 47.67 | 40.00 | 61.33 | **71.67** |
| | 32 | 39.67 | 23.67 | 23.33 | 53.67 | **66.67** |

*Table 4.* Dilated spacing post-filter applied on top of EB-Sampler ($\gamma{=}2$) and CB-Sampler ($\tau{=}0.5$) at $B{=}32$, with a fixed initial gap $g_0{=}8$. Each cell is Acc / NFE. **Bold** indicates accuracy improvement over the unspaced baseline. Full sweep over $(\gamma, \tau, g_0)$ and pseudocode are in Appendix B.10.

| Model | Dataset | EB ($\gamma{=}2$) | | CB ($\tau{=}0.5$) | |
|---|---|---|---|---|---|
| | | off | + spacing | off | + spacing |
| LLaDA-Inst. | HumanEval | 24.4 / 35 | **37.8** / 59 | 22.6 / 22 | **34.8** / 42 |
| | MATH500 | 24.8 / 47 | **27.8** / 81 | 18.4 / 37 | **28.2** / 62 |
| | IFEval | 63.7 / 91 | **65.2** / 132 | 59.7 / 83 | **63.8** / 120 |
| Dream-Inst. | HumanEval | 7.9 / 55 | **9.8** / 101 | 12.8 / 42 | **14.6** / 70 |
| | MATH500 | 26.8 / 47 | **33.6** / 84 | 23.8 / 39 | **31.4** / 69 |
| | IFEval | 46.6 / 87 | **49.0** / 121 | 43.8 / 80 | **47.1** / 108 |

tern provides substantial token spacing while each block still benefits from rich left-context (prompt and previously generated blocks). Very small blocks ($B = 8$) limit DUS because the coarse-to-fine schedule has only $\lceil \log_2 8 \rceil = 3$ levels. Larger blocks ($B = 64+$) require predicting more tokens from sparser within-block context, particularly in early levels where few positions have been revealed, and increase the entropy gap as more tokens are decoded in parallel. Nevertheless, as shown in Sections 4.1-4.2, DUS outperforms confidence-based planners at any fixed $B$; the block size controls the quality-speed operating point. Beyond block size, the remaining knobs ($a = 2$, `base_skip = 1`, threshold $= 0$) are robust across settings (Appendices B.4 and B.2); full single-block ($B{=}G$, $\sim 28\times$ speedup) and wall-clock results are in Appendices B.3 and B.8.

### 4.4. Beyond DUS: Dilated Spacing as a Drop-in Filter for Adaptive Schedulers

Adaptive token-selection schedulers such as the EB-Sampler (Ben-Hamu et al., 2025) and the CB-Sampler (Wu et al., 2025) reach higher accuracy than DUS at higher NFE on some benchmarks (Appendix B.9). They decide how many tokens to unmask per step from a per-token score, but place no constraint on where those tokens are (except the semi-AR block constraint). We ask whether the dilated spacing principle from DUS can improve these strong baselines, while maintaining their core principles.

Given the scheduler's selected candidates $\mathcal{I}$ at step $t$ within the current block of size $B$, we process $\mathcal{I}$ in descending score order and partition it into an accepted set $\mathcal{I}'$ and a rejected set $\mathcal{R}$ (with $\mathcal{I} = \mathcal{I}' \sqcup \mathcal{R}$): each candidate $i$ is accepted into $\mathcal{I}'$ if $\min_{j \in \mathcal{I}'} |i - j| \geq gap$ and otherwise placed in $\mathcal{R}$. Rejected tokens stay masked, and the scheduler reconsiders them at the next step. The minimum gap is adaptive:

$$gap = \max\left(1, \; \lfloor M_{\text{rem}} \cdot g_0 / B \rfloor\right), \tag{13}$$

where $M_{\text{rem}} = B - |\mathcal{U}_t|$ is the count of still-masked positions in the block and $g_0$ is the *initial gap*: when the block is fully masked, $gap = g_0$. Larger $g_0$ enforces sparser early unmasking; as the block fills, $M_{\text{rem}}$ shrinks and the gap relaxes toward 1, recovering the underlying scheduler. Full pseudocode (Algorithm 1) and per-$(\gamma, \tau, g_0)$ sweeps (Figure 6) are in Appendix B.10.

Table 4 reports an aggressive EB-Sampler ($\gamma{=}2$) and CB-Sampler ($\tau{=}0.5$) at $B{=}32$, with and without the post-filter at a fixed $g_0{=}8$, on LLaDA-Instruct-8B and Dream-Instruct-7B across HumanEval, MATH500, and IFEval. Spacing improves accuracy in all cases, with gains of up to $+13.4\%$ for EB on HumanEval and $+12.2\%$ for CB on HumanEval. Aggressive base settings, where the unfiltered scheduler reveals tightly clustered indices, benefit most from the post-filter; the broader sweep over $(\gamma, \tau, g_0)$ in Figure 6 confirms that for every (model, dataset) at least one $g_0$ improves both accuracy and NFE for EB and CB, supporting the view that dilated spacing is complementary to score-based selection.

## 5. Conclusions

We introduced DUS, a purely inference-time, model-free, model-agnostic planner for MDLMs. In extensive experiments on diffusion LLMs, DUS consistently outperforms the traditional denoiser-confidence planner, improving downstream task accuracy by up to 27% on challenging math and code benchmarks, while simultaneously reducing the number of denoising iterations by an order of magnitude. Unlike typical speed-quality trade-offs, our method both accelerates inference and enhances output quality. The dilated-spacing principle is also complementary to per-token adaptive samplers - applied as a post-filter on EB-Sampler and CB-Sampler selections it improves their accuracy at matched or modestly higher NFE (Section 4.4) - and orthogonal to KV-caching for diffusion models (Hu et al., 2025; Wu et al., 2025; Ma et al., 2026), which reduces per-step cost without altering the unmasking schedule. By unlocking the parallelism inherent in diffusion decoding without any modifications to model architecture or training, DUS reveals a new path for inference-only planners to fully exploit MDLM parallelism.

## Impact Statement

This paper presents an inference-time optimization method for masked diffusion language models. By reducing the number of calls required for text generation while maintaining or improving output quality, DUS has the potential to lower computational costs and energy consumption associated with using deployed diffusion-based language models. More efficient inference enables broader access to these models and reduces their environmental footprint. As with any advancement in large language model capabilities, practitioners should remain careful of potential misuse, however, DUS does not introduce new generative capabilities beyond what existing models already provide.

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

# A. Setup and Background

## A.1. Use of Large Language Models

A large language model was used solely for minor editing of draft text. The model was not used for ideation, experimental design, implementation, analysis, data generation, or to produce technical content. All scientific claims and results were produced and verified by the authors.

## A.2. Masked Diffusion: Forward Process and Training Objective

We formally define the forward and reverse processes used during training of masked diffusion language models (Sahoo et al., 2024; Nie et al., 2025b; Ye et al., 2025), and distinguish them from the inference procedure.

**Forward process (training).** Given a clean sequence $\mathcal{X} = (X_1, \ldots, X_G)$ with $X_i \in \mathcal{V}$, the forward process at noise level $t \in (0, 1]$ produces a masked sequence $\mathcal{M}_t = (M_{t,1}, \ldots, M_{t,G})$ by independently replacing each token with [MASK] with probability $t$:

$$M_{t,i} = \begin{cases} X_i & \text{w.p. } 1-t, \\ \text{[MASK]} & \text{w.p. } t. \end{cases} \quad (14)$$

At $t = 0$ the sequence is fully observed; at $t = 1$ all tokens are masked.

**Reverse process (training).** The reverse process recovers the clean sequence from the masked state. The denoiser $\mathcal{D}_\theta$ is trained to reconstruct the original tokens by minimizing a reweighted cross-entropy loss over masked positions:

$$\mathcal{L}(\theta) = -\mathbb{E}_{t\sim U(0,1],\, \mathcal{X},\, \mathcal{M}_t}$$
$$\left[\frac{1}{t}\sum_{i=1}^{G} \mathbf{1}[M_{t,i} = \texttt{[MASK]}]\, \log p_\theta(X_i \mid \mathcal{S}_t)\right],$$

where $\mathcal{S}_t$ denotes the sampler state (the information in $\mathcal{M}_t$). This objective upper-bounds the negative log-likelihood $-\log p_\theta(\mathcal{X})$ (Sahoo et al., 2024; Nie et al., 2025b). The $1/t$ reweighting compensates for fewer masked tokens at low noise levels.

**Inference.** At inference time, $\theta$ is fixed and generation mimics the reverse process: starting from a completely or partially masked state (e.g., an unmasked prompt followed by masked generation tokens), a planner $\mathcal{P}_t$ iteratively selects indices $\mathcal{I}_t$ to unmask, the denoiser predicts token distributions $p_\theta(X_i \mid \mathcal{S}_t)$ for $i \in \mathcal{I}_t$, tokens are sampled, and the state is updated. This paper focuses on the design of the planner $\mathcal{P}_t$.

### A.3. Datasets and Settings

All experiments were conducted on NVIDIA Tesla V100 GPUs. Evaluation was performed using the *Language Model Evaluation Harness* repository (https://github.com/EleutherAI/lm-eval-harness); our code, based on the LLaDA repository (https://github.com/ML-GSAI/LLaDA), is publicly available at https://github.com/omerlux/DUS for reproducibility. Average NFEs per block (and overall for each generation length) are reported to ensure a fair comparison across methods.

For the standard benchmarks GSM8K (Cobbe et al., 2021), MBPP (Austin et al., 2021), and HumanEval (Chen et al., 2021), evaluation is conducted on the full datasets (1,319, 500, and 164 samples, respectively), with GSM8K parallelism ablations limited to the first 300 samples. To facilitate evaluation on MATH500 (Lightman et al., 2023), a new task class is implemented for the 500 cherry-picked samples from the MATH dataset (Hendrycks et al., 2021). Finally, to assess robustness on reasoning and professional-knowledge benchmarks, subsets are sampled from BBH (Suzgun et al., 2023) (20 examples from each of its 27 subgroups, 540 total) and MMLU-Pro (Wang et al., 2024) (40 examples from each of its 14 subgroups, 560 total). Datasets' generation length $G$ is set to 256, except for the coding datasets HumanEval and MBPP, where $G = 512$. For IFEval (Zhou et al., 2023), the generation length is $G = 1024$ for the planner parallelism experiments (Figure 2) and $G = 256$ for the hybrid post-filter evaluation (Section 4.4, Table 4),

to match the math/code protocol in the latter case.

| Formula | Definition |
|---|---|
| $G$ | Generation length. |
| $B$ | Block size. |
| $k$ | Average number of unmasked tokens per iteration. |
| $n_{\text{blocks}} = \dfrac{G}{B}$ | Number of blocks. |
| $\text{NFE}_{\text{block}} = \dfrac{B}{k}$ | Number of function evaluations per block. |
| $\text{NFE} = n_{\text{blocks}} \cdot \text{NFE}_{\text{block}}$ | Total number of function evaluations. |
| $\dfrac{\text{NFE}}{\text{NFE}_{\text{block}}} = \dfrac{B}{k}$ | NFE speedup factor compared to token-by-token inference. |

*Table 5.* Inference NFE speedup conversion table.

All evaluations use a few-shot, COT prompting framework: GSM8K and MBPP with 4-shot contexts, HumanEval with 0-shot, MATH500 with 4-shot, BBH with 3-shot, and MMLU-Pro with 5-shot.

Early stop for generation is implemented for all planners, which ends generation if the $\texttt{[EOS]}$ token is unmasked and all previous tokens are unmasked too, since generating text after this token would produce content unrelated to the answer.

Table 5 summarizes the notation and formulas used throughout this paper. In most experiments, the unmasking parameter is set as $k = \log_2 B$ unless otherwise specified.

## B. Experiments and Ablations

### B.1. Block-Size Effect on Speedups

In this experiment, the effect of block size $B$ on generation accuracy was evaluated under a fixed total NFEs, corresponding to an $8\times$ speedup relative to token-by-token decoding. The DUS was configured to begin at a higher iteration $t_0 > 1$, resulting in larger unmasking group sizes $k$ per step (cf. (8)). Block sizes $B = \{8, 16, 32\}$, corresponding to $\text{NFE}_{\text{block}} = \{1, 2, 4\}$, were tested on GSM8K (first 300 samples). Table 6 reports task accuracy (%) for LLaDA-Base and LLaDA-Instruct, revealing a monotonic increase in performance with $B$: the Base model rose from 13.33% at $B = 8$ to 32.33% at $B = 32$ ($\approx 2.4\times$ accuracy improvement), while the Instruct variant climbed from 12.00% to 56.67% ($\approx 4.7\times$ accuracy improvement).

These gains are attributed to the fact that larger block sizes, when combined with DUS, spread the initially predicted tokens farther apart. This spatial separation reduces MI

*Table 6.* Speedup comparison on GSM8K (300 samples, $G = 256$) using DUS under an $8\times$ inference budget (total NFE $= 32$). Block sizes $B \in \{8, 16, 32\}$ correspond to average NFEs per block of $\{1, 2, 4\}$. Results for both base and instruct LLaDA models; best scores are **bold** and second-best are underlined.

| Block Size | 8 | 16 | 32 |
|---|---|---|---|
| **Avg NFEs@Block** | 1 | 2 | 4 |
| **Model** | | Score (%, ↑) | |
| Base | 13.33 | 11.00 | **32.33** |
| Instruct | 12.00 | 16.00 | **56.67** |

among unmasked tokens in early iterations - consistent with our analysis in Lemma 3.6 - and allows subsequent iterations to fill in the gaps and more effectively correct existing errors introduced by coarse-grained parallelism. Tuning $B$ thus offers an additional lever to boost output quality without raising the compute budget, though excessively large $B$ may force the model to predict tokens with insufficient nearby context, which can exceed the model capabilities.

## B.2. Skip Threshold Ablation

The skip heuristic introduced in Section 3.5 defers unmasking positions whose denoiser confidence is below a threshold $c$, leaving them masked for reconsideration at the next iteration. Table 7 sweeps $c \in \{0.0, 0.1, 0.3, 0.5\}$ on LLaDA-Base and LLaDA-Instruct, on MATH500 ($B=16$) and HumanEval ($B=32$).

*Table 7.* Skip-DUS ablation: confidence threshold $c$ controls how aggressively DUS defers low-confidence tokens. Each cell is Acc / NFE; $c=0$ recovers standard DUS (matches Tables 1–2). MATH500 uses $B=16$, $G=256$; HumanEval uses $B=32$, $G=512$. **Bold** marks the best score per row.

| Model | Dataset | \multicolumn{4}{c}{Skip threshold $c$} |
|---|---|---|---|---|---|

| Model | Dataset | 0.0 | 0.1 | 0.3 | 0.5 |
|---|---|---|---|---|---|
| LLaDA-Base | MATH500 | 16.8 / 36 | 17.0 / 37 | 17.2 / 39 | **17.6 / 42** |
| | HumanEval | 9.8 / 43 | 9.8 / 37 | **12.8 / 31** | 11.6 / 37 |
| LLaDA-Inst | MATH500 | 21.0 / 38 | **21.2 / 38** | 20.6 / 40 | 20.6 / 46 |
| | HumanEval | 13.4 / 26 | 14.0 / 25 | 9.2 / 23 | **15.2 / 34** |

The default $c=0$ (no skip) is competitive across all four cells, and skip improves the best score in three of them. The most pronounced gain is on HumanEval with LLaDA-Base, where $c=0.3$ both raises pass@1 from $9.8\%$ to $12.8\%$ and lowers NFE from $43$ to $31$. On MATH500 the threshold shifts NFE upward in exchange for modest accuracy gains, while on HumanEval-Instruct $c=0.5$ is best despite higher NFE. Optimal $c$ is therefore task-dependent; the default $c=0$ remains a safe baseline to guarantee speedups.

## B.3. Single-Block DUS ($B=G$)

A natural concern is whether DUS still works in the limit where each block spans the entire generation: $B=G$. With

$G \in \{256, 512\}$ this reduces the schedule to $\lceil \log_2 G \rceil$ denoiser iterations, yielding a substantial NFE speedup. Table 8 sweeps the skip threshold $c \in \{0.0, 0.15, 0.3, 0.45\}$ on LLaDA-Base and LLaDA-Instruct across the three datasets.

*Table 8.* Single-block DUS at $B=G$ - i.e., the entire generation is one dilated block. Each cell is accuracy (%) at the given skip threshold $c$. NFE is $\sim 8$ for $G=256$ (MATH500, GSM8K) and $\sim 10$ for $G=512$ (HumanEval), corresponding to a $\times 28$ and $\times 51$ relative NFE speedup over token-by-token. **Bold** marks the best $c$ per row.

| Model | Dataset | \multicolumn{4}{c}{Skip threshold $c$} | Speedup |
|---|---|---|---|---|---|---|

| Model | Dataset | 0.0 | 0.15 | 0.3 | 0.45 | Speedup |
|---|---|---|---|---|---|---|
| LLaDA-Base | MATH500 | **8.2** | 6.4 | 7.0 | 4.4 | $\times 28$ |
| | GSM8K | 12.5 | 15.9 | **17.4** | 16.8 | $\times 28$ |
| | HumanEval | 9.8 | 8.5 | **11.0** | 9.8 | $\times 51$ |
| LLaDA-Inst | MATH500 | **17.2** | 11.8 | 10.4 | 7.8 | $\times 28$ |
| | GSM8K | **42.3** | 41.9 | 37.9 | 33.1 | $\times 28$ |
| | HumanEval | 14.6 | 14.6 | **16.5** | 14.6 | $\times 51$ |

Single-block DUS produces non-trivial accuracy at all settings, including $42.3\%$ on GSM8K and $17.2\%$ on MATH500 for LLaDA-Instruct at $\times 28$ speedup. The skip threshold $c$ improves the best-row accuracy in three of six (model, dataset) settings, with the largest gain on GSM8K-Base ($12.5\% \rightarrow 17.4\%$). For well-calibrated regimes (LLaDA-Instruct on MATH500 and GSM8K), $c=0$ is best, consistent with the broader skip ablation in Appendix B.2. Overall, the $B=G$ regime trades accuracy for an order-of-magnitude further speedup beyond the $B \in \{8, 16, 32, 64\}$ operating points in Tables 1-2.

## B.4. Parameter Sensitivity

We sweep the two remaining DUS knobs not covered by Tables 1-2 (which span $B \in \{8, 16, 32, 64\}$) or Appendix B.2 (which sweeps the skip threshold $c$): the dilation base $a$ and the level-skip parameter `base_skip`. On LLaDA-Base across MATH500 and GSM8K at $B \in \{16, 32\}$, Table 9 reports the default operating point ($a=2$, `base_skip=1`) alongside two alternative bases ($a \in \{3, 4\}$) and the one alternative level-skip (`base_skip=2`).

*Table 9.* Sensitivity of DUS to the remaining design choices on LLaDA-Base. Each is Acc / NFE. The **Default** column ($a=2$, `base_skip=1`) reproduces Tables 1-2; the other columns vary one parameter at a time. Default settings consistently better.

| Dataset | $B$ | **Default** $a=2$, skip=1 | Base $a$ $a=3$ | $a=4$ | `base_skip` $= 2$ |
|---|---|---|---|---|---|
| MATH500 | 16 | **19.2 / 64** | 6.4 / 22 | 4.4 / 22 | 6.0 / 48 |
| | 32 | **13.6 / 40** | 6.2 / 17 | 4.0 / 20 | 7.8 / 32 |
| GSM8K | 16 | **59.5 / 64** | 26.3 / 23 | 29.3 / 32 | 38.7 / 48 |
| | 32 | **49.4 / 40** | 26.6 / 19 | 23.2 / 24 | 33.4 / 32 |

The default settings dominate at every (dataset, $B$) pair. Raising $a$ from 2 to 3 or 4 reduces the number of dilated

*Table 10.* Per-step runtime breakdown (1 NFE): model forward pass and component overheads (DUS, confidence calculation, CB selection, entropy calculation, EB selection), in milliseconds.

| Block size $B$ | Steps $\log_2(B)$ | Model (ms) | DUS (ms) | Conf calc (ms) | CB sel (ms) | Entr calc (ms) | EB sel (ms) |
|---|---|---|---|---|---|---|---|
| 8 | 4 | 247.34 | 0.0002 | 1.1630 | 0.1524 | 1.2643 | 0.1009 |
| 16 | 5 | 247.34 | 0.0002 | 1.1656 | 0.2775 | 1.3289 | 0.1089 |
| 32 | 6 | 247.34 | 0.0002 | 1.1698 | 0.4307 | 1.4720 | 0.1259 |
| 64 | 7 | 247.34 | 0.0003 | 1.2578 | 0.7213 | 2.0095 | 0.0902 |

levels from $\lceil \log_2 B \rceil$ to $\lceil \log_a B \rceil$, packing far more tokens into each level and breaking the coarse-to-fine structure; accuracy drops by up to $-77\%$ relative on MATH500 and $-56\%$ relative on GSM8K. Raising `base_skip` from 1 to 2 removes the coarsest dilated level so the schedule starts at the next-finer spacing, saving $\approx 25\%$ NFE at the cost of $-32\%$ to $-69\%$ relative accuracy across datasets. These results justify $a{=}2$ and `base_skip=1` as the practitioner defaults referenced in Section 4.3.

## B.5. Speed-Quality Trade-off Analysis

Figure 3 presents the full speed-quality trade-off analysis across GSM8K, HumanEval, and MBPP for all tested models. The plots show accuracy vs. speedup for $2.7\times$, $4\times$, $6.4\times$, and $10.7\times$ acceleration (block sizes $B \in \{8, 16, 32, 64\}$). Smaller blocks yield higher accuracy at the cost of more denoiser rounds (higher NFE), while larger blocks enable up to $10\times$ fewer iterations but lower end accuracy. Across all datasets and speedup factors, DUS consistently achieves higher scores compared to the self-confidence planner. GSM8K shows a steady decline in scores as inference speed increases, whereas the code benchmarks exhibit less smooth trends, likely reflecting their already low performance at higher speedups. Nonetheless, DUS delivers more consistent improvements across all settings.

## B.6. Runtime Breakdown: Sampler Overhead vs. Denoiser

We report a per-step (1 NFE) runtime breakdown decomposing the cost into the *model forward pass* and individual *sampler component overheads*. Each adaptive sampler combines a scoring step and a selection step: the CB-Sampler uses confidence calculation + CB selection, while the EB-Sampler uses entropy calculation + EB selection. DUS requires only a lightweight schedule lookup. All timings are in milliseconds, measured on LLaDA-8B-Base with a single RTX 6000 Ada GPU across block sizes $B \in \{8, 16, 32, 64\}$.

As shown in Table 10, the scheduler selection step is negligible across block sizes (DUS selection is $2.6 \times 10^{-4}$-$3.1 \times 10^{-4}$ ms per step). The dominant sampler costs come from scoring: confidence computation is $\approx 1.16$-$1.26$ ms per step, while entropy computation is higher and grows

with $B$ ($\approx 1.26$-$2.01$ ms). CB selection also incurs a non-trivial overhead that increases with $B$ ($\approx 0.15$-$0.72$ ms), whereas EB selection remains relatively small. Overall, all sampler components remain small compared to the model forward pass ($247.34$ ms per step).

## B.7. Token Distance Analysis Across Planners

We measure the spatial pattern produced by each planner directly from generation traces. For each generation, we extract the set of token positions revealed at every denoiser iteration and compute (i) the average pairwise distance between simultaneously revealed tokens and (ii) the fraction of revealed tokens that are isolated (no adjacent co-decoded token). Table 11 reports these metrics on LLaDA-8B-Base across MATH500 and HumanEval at block sizes $B \in \{16, 32\}$, comparing DUS, self-confidence (SC), entropy-bounded (EB, $\gamma = 1.0$) and confidence-bounded (CB, $\tau = 0.7$) planners.

DUS attains $2$-$3\times$ wider average spacing than every alternative and isolates essentially all co-decoded tokens; SC clusters most aggressively, with 49-58% of co-decoded tokens lying adjacent to another co-decoded token. This clustering arises because neighboring masked tokens often share similar context and therefore similar conditional entropy or confidence; denoiser-guided self-planners consequently reveal multiple nearby tokens in the same step while leaving large masked regions elsewhere. Although these planners minimize the sum of marginal entropies (cf. (5)), they can maintain higher overall uncertainty across iterations, since selecting the *easiest* (lowest-entropy) tokens first postpones the *hard* tokens, which remain both (i) high-entropy and (ii) weakly supported by newly revealed context. The pattern thus validates the schedule's spacing guarantee on tasks (code, reasoning) that violate strict fast mixing, supporting the discussion in Section 3.6.

DUS breaks this feedback loop by enforcing spatially distributed reveals via the dilated pattern, improving context propagation across the block: after each iteration, many remaining masked tokens gain at least some nearby revealed context from multiple regions, which tends to reduce their entropies more uniformly in subsequent steps. Figure 4 illustrates this on a single block at $B = 16$, comparing entropy (incremental $k$), entropy (fixed $k$), and DUS (dilated, position-based).

## B.8. Wall-Clock Throughput

We complement the per-step microbenchmark of the previous subsection with end-to-end wall-clock generation time across all schedulers, on a fixed model/dataset/hardware setup (LLaDA-8B-Base, GSM8K, RTX 6000 Ada). Table 12 reports the median per-sample wall-clock alongside empirical NFE; the two quantities scale together because the

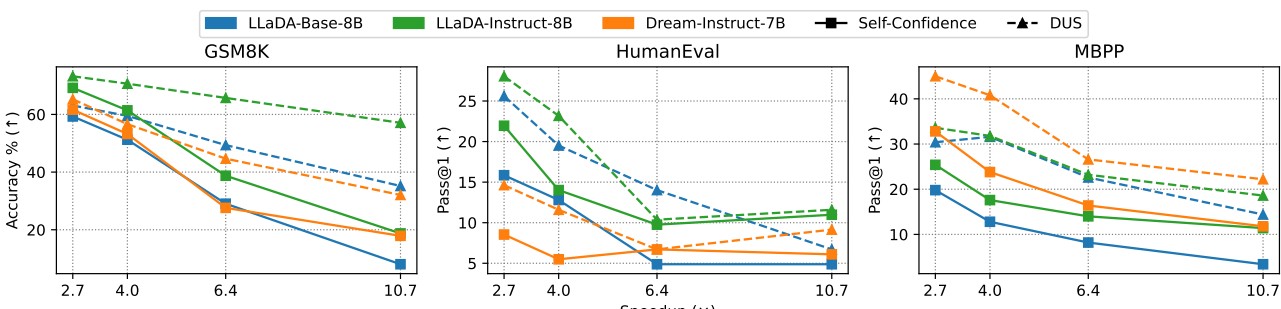

*Figure 3.* Speed-quality trade-off on GSM8K, HumanEval, and MBPP for various speedup factors defined by semi-AR inference block size $B \in \{8, 16, 32, 64\}$. Higher score (Accuracy / Pass@1) is better. Each color represents a different model; markers indicate planners: self-confidence (■) and DUS (▲).

*Table 11.* Spatial pattern produced by each planner on LLaDA-8B-Base (150 samples per row). Left half: average pairwise distance between co-decoded tokens within a block. Right half: percentage of isolated tokens (no adjacent co-decoded token). DUS spreads simultaneously revealed tokens 2-3× wider than the alternatives, while almost every co-decoded token is isolated.

| Dataset | $B$ | Avg Pairwise Distance | | | | % Isolated Tokens | | | |
|---|---|---|---|---|---|---|---|---|---|
| | | DUS | SC | EB | CB | DUS | SC | EB | CB |
| MATH500 | 16 | **6.22** | 3.00 | 3.49 | 4.47 | **100** | 51.0 | 64.8 | 70.7 |
| | 32 | **9.62** | 4.13 | 4.74 | 7.13 | **100** | 49.8 | 66.0 | 71.1 |
| HumanEval | 16 | **6.22** | 2.81 | 4.37 | 4.71 | **100** | 49.0 | 62.0 | 70.1 |
| | 32 | **11.59** | 3.71 | 6.39 | 8.63 | **100** | 42.1 | 66.2 | 70.0 |

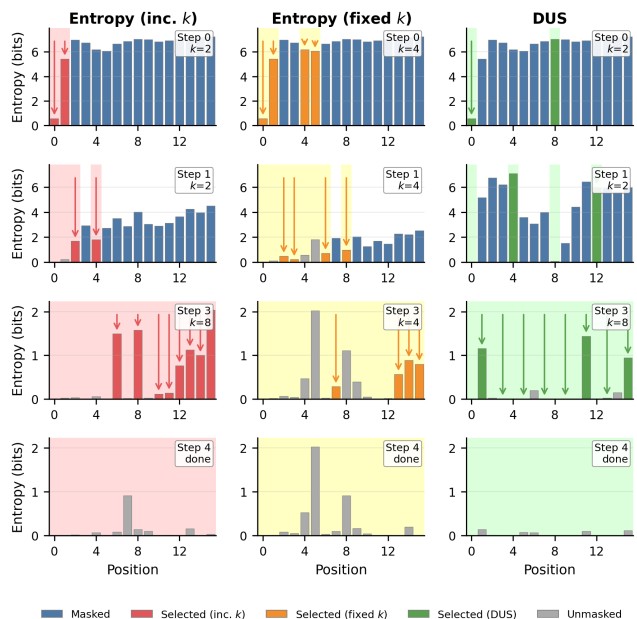

*Figure 4.* Token selection strategies on a single block. Entropy-based methods (left, center) greedily select low-entropy tokens, leading to spatial clustering and persistent uncertainty in remaining tokens. DUS (right) uses position-based selection, yielding more uniform uncertainty reduction across the sequence.

model forward pass dominates per-step cost (cf. Table 10). At $B = 16$, DUS achieves a $3.9\times$ wall-clock speedup over token-by-token decoding while reaching $59.51\%$ accuracy on GSM8K; at $B = 32$, the speedup grows to $5.8\times$ at $49.36\%$. As a reference point, LLaDA-8B with DUS ($B = 16$) is $5.6\times$ faster than the autoregressive Llama-3-8B baseline on the same hardware.

## B.9. Adaptive Scheduler Comparison

We additionally compare DUS against two adaptive token-selection samplers from the literature: the entropy-bounded sampler (EB-Sampler) (Ben-Hamu et al., 2025), which unmasks the largest set of tokens whose cumulative conditional entropy is below a user-specified bound $\gamma$, and the confidence-bounded sampler (CB-Sampler) (Wu et al., 2025), which unmasks every token whose confidence exceeds a threshold $\tau$. Both methods are inference-only and adaptive: the number of tokens unmasked per step varies per sample.

Figure 5 plots accuracy versus empirical NFE across all 5 MDLMs and 4 benchmarks. DUS sweeps over $B \in \{8, 16, 32, 64\}$; EB-Sampler over $\gamma \in \{4.0, 1.0, 0.1, 0.01\}$ (extended for DiffuCoder); CB-Sampler over $\tau \in \{0.5, 0.7, 0.9\}$ (extended for DiffuCoder). DUS sits at the lowest-NFE end of every panel, providing the most aggressive speedup at the cost of some accuracy; EB-Sampler and CB-Sampler reach higher accuracy at higher NFE, with task-dependent ordering.

*Table 12.* End-to-end wall-clock generation time on a single NVIDIA RTX 6000 Ada (LLaDA-8B-Base, GSM8K, generation length 256; 105 samples after 5 warmup). Median per-sample wall-clock, empirical NFE, speedup vs. token-by-token, and full-dataset GSM8K accuracy from Tables 1-2. The model forward pass dominates per-step cost ( 99%), so wall-clock tracks NFE.

| Method | Config | Median (ms) | NFE | Speedup | GSM8K (%) |
|---|---|---|---|---|---|
| Llama-3-8B (AR) | - | 35,144 | 257 | 0.69× | 49.81 |
| Token-by-token | $B=256$ | 24,280 | 124 | 1× | 72.63 |
| Self-confidence | $B=16$ | 5,783 | 32.9 | 4.2× | 51.23 |
| **DUS** | $B=16$ | 6,314 | 32.8 | 3.9× | 59.51 |
| **DUS** | $B=32$ | 4,182 | 22.8 | 5.8× | 49.36 |
| EB ($\gamma=1.0$) | $B=16$ | 8,577 | 43.3 | 2.8× | 71.72 |
| CB ($\tau=0.7$) | $B=16$ | 7,088 | 38.3 | 3.4× | 72.02 |

## B.10. Dilated Spacing Post-Filter Details

This appendix gives the full pseudocode (Algorithm 1) of the dilated spacing post-filter introduced in Section 4.4, together with per-$(\gamma, \tau, g_0)$ sweeps (Figure 6) that complement the summary in Table 4.

---

**Algorithm 1** Dilated Spacing Post-Filter

---

**Require:** block size $B$; selected indices $\mathcal{I}$ (sorted by score, descending); unmasked set $\mathcal{U}_t$ and masked set $\mathcal{M}_t$ in the block (with $\mathcal{I} \subseteq \mathcal{M}_t$); initial gap $g_0$
1: $gap \leftarrow \max(1, \lfloor |\mathcal{M}_t| \cdot g_0/B \rfloor)$
2: $\mathcal{I}' \leftarrow \emptyset; \quad \mathcal{R} \leftarrow \emptyset$
3: **for** $i \in \mathcal{I}$ (high-score first) **do**
4:     **if** $\min_{j \in \mathcal{I}'} |i - j| \geq gap$ **then**
5:         $\mathcal{I}' \leftarrow \mathcal{I}' \cup \{i\}$
6:     **else**
7:         $\mathcal{R} \leftarrow \mathcal{R} \cup \{i\}$
8:     **end if**
9: **end for**
10: $\mathcal{M}_t \leftarrow (\mathcal{M}_t \setminus \mathcal{I}) \cup \mathcal{R}$
11: **return** $\mathcal{I}'$

---

Figure 6 reports the full sweep across initial gaps $g_0 \in \{\text{off}, 4, 8, 16\}$, base parameters ($\gamma \in \{1, 2, 4\}$ for EB; $\tau \in \{0.3, 0.5, 0.7\}$ for CB), 2 instruct models and 3 datasets. Because rejected indices stay masked, the post-filter generally requires equal or slightly more denoiser iterations than the base scheduler at the same $(\gamma, \tau)$; in most cases this additional NFE translates into higher accuracy, with the largest gains for aggressive base parameters (large $\gamma$, low $\tau$) where the unfiltered scheduler reveals tightly clustered indices. Performance is not strictly monotone in $g_0$, with occasional regressions on Dream-Instruct CB. Overall the post-filter exposes a usable trade-off where additional NFE is converted into accuracy, leaving the underlying scheduler's score function and selection criterion unchanged.

## C. Theory Guarantees

### C.1. Surrogate-to-Joint Entropy Gap Decomposition

This appendix provides the full statement and proof for Lemma 3.3.

**Lemma 3.3 (Token-by-token vs. parallel bounds) (restated).** *Let the current block indices be* $\mathcal{B} := \{b, \ldots, b + B - 1\}$. *Consider any partition of the block into* $T \in \mathbb{N}$ *disjoint groups* $\mathcal{I}_1, \ldots, \mathcal{I}_T$ *with* $\bigcup_{t=1}^{T} \mathcal{I}_t = \mathcal{B}$, *and define* $\mathcal{I}_{<t} := \bigcup_{s<t} \mathcal{I}_s$. *Under the optimal estimator assumption, the total loss induced by parallel sampling within each group is* $\sum_{t=1}^{T} \mathcal{L}(\mathcal{I}_t; \mathcal{S}_1, X_{\mathcal{I}_{<t}})$. *Then*

$$H(X_\mathcal{B} \mid \mathcal{S}_1) \leq \sum_{t=1}^{T} \mathcal{L}(\mathcal{I}_t; \mathcal{S}_1, X_{\mathcal{I}_{<t}}) \leq \sum_{i \in \mathcal{B}} H(X_i \mid \mathcal{S}_1). \quad (15)$$

*Proof of Lemma 3.3.* Fix a block $\mathcal{B} := \{b, \ldots, b + B - 1\}$, a state $\mathcal{S}_1$, and a partition $\mathcal{I}_1, \ldots, \mathcal{I}_T$ of $\mathcal{B}$, and write $\mathcal{I}_{<t} := \bigcup_{s<t} \mathcal{I}_s$. By the chain rule for entropy,

$$H(X_\mathcal{B} \mid \mathcal{S}_1) = \sum_{t=1}^{T} H(X_{\mathcal{I}_t} \mid \mathcal{S}_1, X_{\mathcal{I}_{<t}}). \quad (16)$$

For each group $\mathcal{I}_t = \{i_1^{(t)}, \ldots, i_{k_t}^{(t)}\}$, applying the chain rule again gives

$$\begin{aligned}
&H(X_{\mathcal{I}_t} \mid \mathcal{S}_1, X_{\mathcal{I}_{<t}}) \\
&= \sum_{j=1}^{k} H(X_{i_j} \mid \mathcal{S}_1, X_{\mathcal{I}_{<t}}, X_{i_1}, \ldots, X_{i_{j-1}}) \\
&\leq \sum_{j=1}^{k} H(X_{i_j} \mid \mathcal{S}_1, X_{\mathcal{I}_{<t}}), \quad (17)
\end{aligned}$$

where the inequality uses that conditioning reduces entropy. Summing over $t$ and using (16) yields the lower bound

$$\begin{aligned}
H(X_\mathcal{B} \mid \mathcal{S}_1) &\leq \sum_{t=1}^{T} \sum_{i \in \mathcal{I}_t} H(X_i \mid \mathcal{S}_1, X_{\mathcal{I}_{<t}}) \\
&= \sum_{t=1}^{T} \mathcal{L}(\mathcal{I}_t; \mathcal{S}_1, X_{\mathcal{I}_{<t}}). \quad (18)
\end{aligned}$$

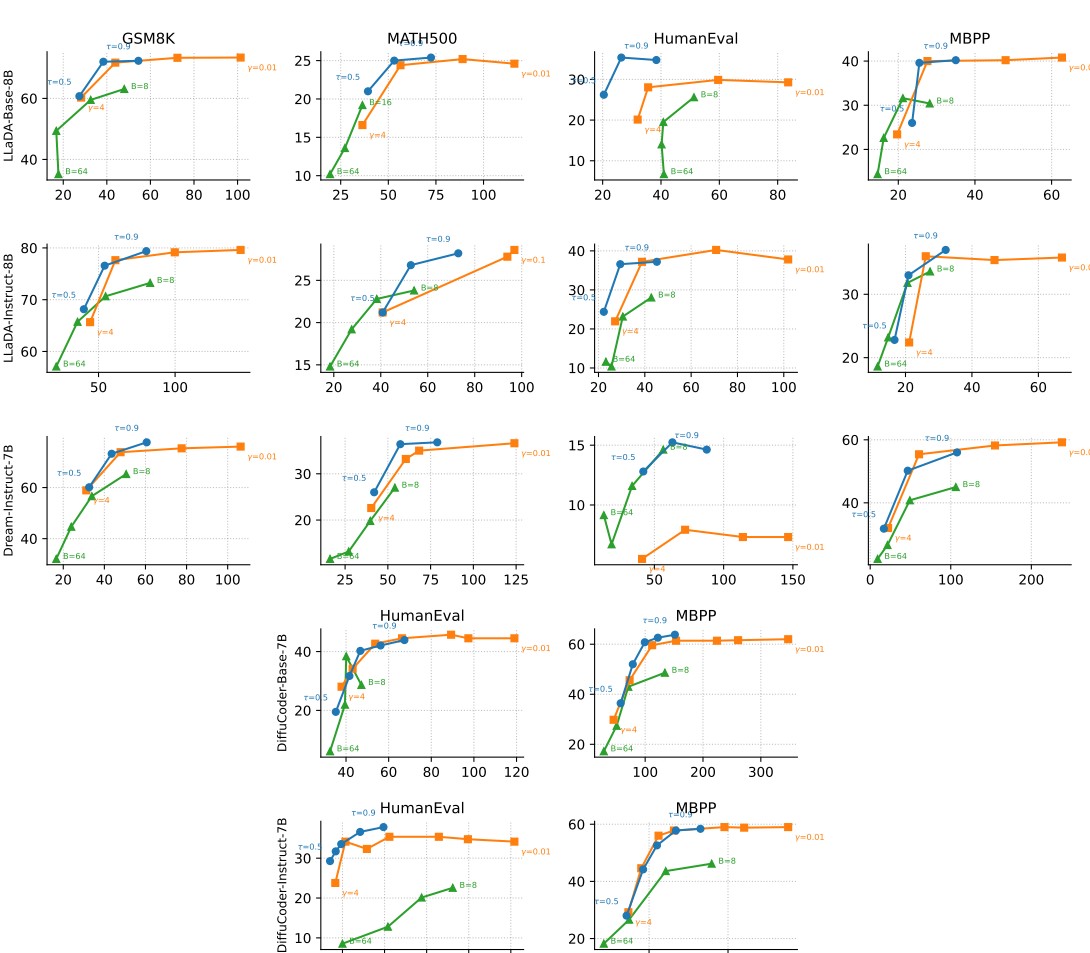

*Figure 5.* Accuracy vs. empirical NFE across 5 MDLMs and 4 benchmarks for DUS (green), EB-Sampler (orange), and CB-Sampler (blue). Parameter extremes are annotated next to the corresponding markers (e.g., $B$=64 and $B$=8 for DUS, $\gamma$=4 and $\gamma$=0.01 for EB, $\tau$=0.5 and $\tau$=0.9 for CB). DiffuCoder rows are code-only.

For the upper bound, again by conditioning reducing entropy, $H\big(X_i \mid \mathcal{S}_1, X_{\mathcal{I}_{<t}}\big) \leq H\big(X_i \mid \mathcal{S}_1\big)$ for all $i$ and $t$, so

$$\sum_{t=1}^{T} \mathcal{L}\big(\mathcal{I}_t; \mathcal{S}_1, X_{\mathcal{I}_{<t}}\big) \leq \sum_{t=1}^{T} \sum_{i \in \mathcal{I}_t} H\big(X_i \mid \mathcal{S}_1\big)$$

$$= \sum_{i \in \mathcal{B}} H\big(X_i \mid \mathcal{S}_1\big), \qquad (19)$$

which completes the proof. $\qquad\square$

### C.2. Lemma 3.5 - Restate and proof

**Lemma 3.5 (restated).** *Assume that $(X_1, \ldots, X_G)$ is a stationary, ergodic VLMC with finite order $L$ with fast-mixing property. Let $1 \leq i_1 < \cdots < i_k \leq B$ be indices*

*such that the pairwise distances satisfy $|i_m - i_n| \geq d$ for all $m \neq n$. Then, for any $\varepsilon > 0$, there exists a distance threshold $D_\varepsilon$ such that for all $d \geq D_\varepsilon$,*

$$H\big(X_{i_1}, \ldots, X_{i_k} \mid \mathcal{S}_t\big) \geq \sum_{j=1}^{k} H\big(X_{i_j} \mid \mathcal{S}_t\big) - \varepsilon. \quad (20)$$

*Proof of Lemma 3.5.* Fix $\varepsilon > 0$. By Lemma 3.6, there exists $D_\varepsilon$ such that for any pair of indices $(i_1, \ldots, i_k)$ with $|i_m - i_n| \geq D_\varepsilon$ for all $m \neq n$, and any realization of $\mathcal{S}_t$,

$$I\big(X_{i_k}; X_{i_1}, \ldots, X_{i_{k-1}} \mid \mathcal{S}_t\big) \leq \frac{\varepsilon}{(k-1)^2}. \quad (21)$$

Assume $d \geq D_\varepsilon$, so that (21) holds for all $m \neq n$.

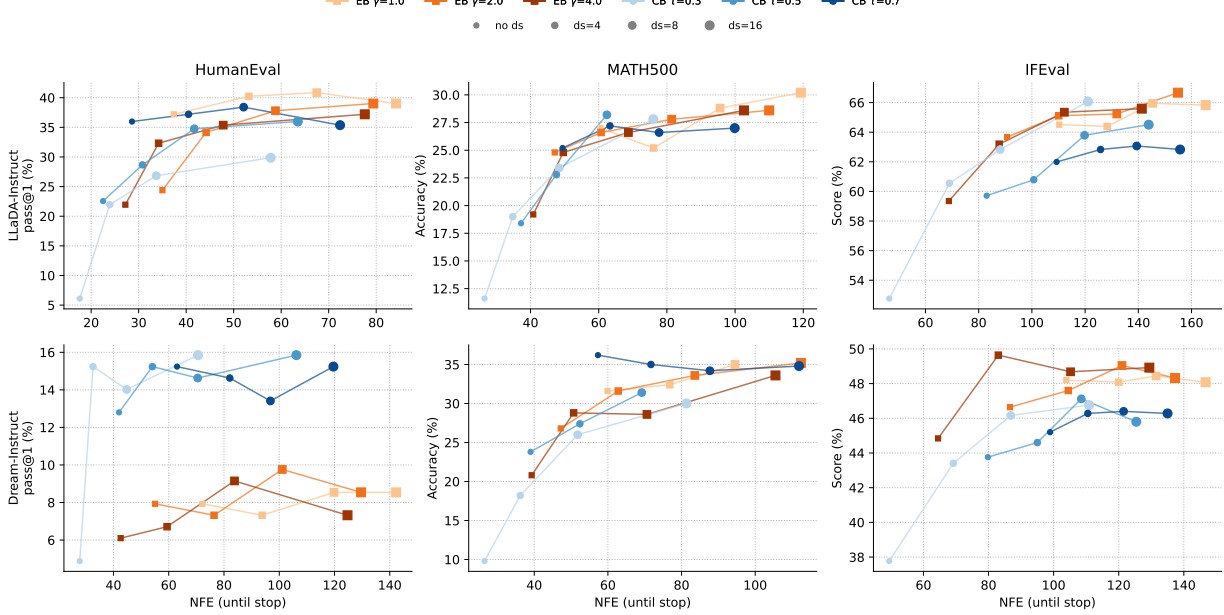

*Figure 6.* EB-Sampler and CB-Sampler with and without the dilated spacing post-filter, on LLaDA-Instruct-8B and Dream-Instruct-7B across HumanEval, MATH500, and IFEval ($B{=}32$). Color shades encode the base parameter ($\gamma$ for EB, $\tau$ for CB; lighter to darker as the parameter increases); marker size encodes the initial gap $g_0 \in \{\text{off}, 4, 8, 16\}$. Each line traces a single (scheduler, $\gamma$ or $\tau$) operating point as $g_0$ increases.

Without loss of generality, let $i_1 < \cdots < i_k$ be the indices of the $k$ tokens chosen in this iteration, each pair separated by at least $D_\varepsilon$. The joint conditional entropy decomposes as

$$H(X_{i_1}, \ldots, X_{i_k} \mid \mathcal{S}_t) = \sum_{j=1}^{k} H(X_{i_j} \mid X_{i_1}, \ldots, X_{i_{j-1}}, \mathcal{S}_t)$$

$$= H(X_{i_1} \mid \mathcal{S}_t) +$$
$$\sum_{j=2}^{k} \left[ H(X_{i_j} \mid \mathcal{S}_t) - I(X_{i_j}; X_{i_1}, \ldots, X_{i_{j-1}} \mid \mathcal{S}_t) \right]$$

$$= \sum_{j=1}^{k} H(X_{i_j} \mid \mathcal{S}_t) - \sum_{j=2}^{k} I(X_{i_j}; X_{i_1}, \ldots, X_{i_{j-1}} \mid \mathcal{S}_t).$$

By Corollary 3.4, the sum of conditional mutual informations above is exactly the entropy gap $\Delta(\mathcal{I}_t; \mathcal{S}_t)$ between the factorized surrogate $\sum_{j=1}^{k} H(X_{i_j} \mid \mathcal{S}_t)$ and the joint objective $H(X_{\mathcal{I}_t} \mid \mathcal{S}_t)$:

$$\Delta(\mathcal{I}_t; \mathcal{S}_t) = \sum_{j=2}^{k} I\left(X_{i_j}; X_{i_1}, \ldots, X_{i_{j-1}} \mid \mathcal{S}_t\right). \quad (22)$$

Thus, minimizing this gap yields an $\varepsilon$-tight approximation of the joint entropy by the marginal-entropy surrogate. By (21),

$$I(X_{i_j}; X_{i_1}, \ldots, X_{i_{j-1}} \mid \mathcal{S}_t)$$
$$\leq (j-1) \cdot \frac{\varepsilon}{(k-1)^2} \leq \frac{\varepsilon}{k-1}.$$

Plugging this into the entropy decomposition above, with $k-1$ bounded terms of MI gives

$$H(X_{i_1}, \ldots, X_{i_k} \mid \mathcal{S}_t) \geq \sum_{j=1}^{k} H(X_{i_j} \mid \mathcal{S}_t) - \varepsilon, \quad (23)$$

which is exactly the claimed inequality (10). □

### C.3. MI Decay Under Fast Mixing

This appendix justifies Lemma 3.6, which bounds the MI between a token and a sparse block of future tokens under the fast-mixing VLMC assumption.

**Assumption C.1** (Fast mixing VLMC). *Let $(X_i)_{i \in \mathbb{Z}}$ be a VLMC with finite maximal context $L < \infty$ on a finite alphabet. Assuming the induced Markov chain on contexts of length $L$ is irreducible and aperiodic, which implies a unique stationary distribution and geometric mixing.*

**Lemma 3.6 (MI decays under fast mixing) (restated).**
*Under Assumption C.1, there exist constants $C < \infty$ and $\rho \in (0, 1)$ such that for any index $i$, spacing $k \geq 1$, and any finite $M \geq 0$,*

$$I\left(X_i; X_{i+k}, X_{i+2k}, \ldots, X_{i+(M+1)k}\right) \leq C\rho^k. \quad (24)$$

Note that the distance between $X_i$ and any symbol in the future block is at least $k$.

*Proof.* Define contexts of length $L$ as $C_i = (X_i, X_{i+1}, \ldots, X_{i+(L-1)}) \in \mathcal{C}$. The sequence $(C_i)$ forms a finite-state Markov chain which is irreducible and aperiodic, implying geometric mixing. Observe that a block of $M+1$ future symbols is a deterministic function of the future contexts,

$$(X_{i+k}, X_{i+2k}, \ldots, X_{i+(M+1)k}) = \\ f(C_{i+k}, C_{i+2k}, \ldots, C_{i+(M+1)k}). \quad (25)$$

The data processing inequality (DPI) states that if $Y = f(Z)$ is a function of $Z$, then $I(X;Y) \leq I(X;Z)$. Letting $Y = (X_{i+k}, X_{i+2k}, \ldots, X_{i+(M+1)k})$ and $Z = (C_{i+k}, C_{i+2k}, \ldots, C_{i+(M+1)k})$, it follows that

$$I(X_i; X_{i+k}, X_{i+2k}, \ldots, X_{i+(M+1)k}) \\ \leq I(X_i; C_{i+k}, C_{i+2k}, \ldots, C_{i+(M+1)k}). \quad (26)$$

Since $X_i$ is a function of the current context $C_i$, applying DPI again yields

$$I(X_i; C_{i+k}, C_{i+2k}, \ldots, C_{i+(M+1)k}) \\ \leq I(C_i; C_{i+k}, C_{i+2k}, \ldots, C_{i+(M+1)k}).$$

Thus, it suffices to show that the MI between the current context $C_i$ and a block of future contexts decays with the separation $k$.

Since $(C_i)$ is finite-state, irreducible, and aperiodic, there exists $\alpha < \infty$ and $0 < \rho < 1$ such that for any state $c$ the total variation (TV) between the stationary and conditioned on past probabilities are bounded by an exponentially decaying factor dependent on $k$ (Bradley, 2005),

$$\|P(C_{i+k} \mid C_i = c) - \pi\|_{TV} \leq \alpha \rho^k. \quad (27)$$

For a block of $M+1$ future contexts, standard finite-state Markov chain bounds imply

$$\|P(C_{i+k}, C_{i+2k}, \ldots, C_{i+(M+1)k} \mid C_i = c) - \pi^{\otimes(M+1)}\|_{TV} \\ \leq (M+1)\alpha\rho^k.$$

The MI can be bounded in terms of TV distance (Zhang, 2007). In particular, if $\|P_{AB} - P_A \otimes P_B\|_{TV} \leq \delta$, then $I(A;B) \leq \beta\delta \log(1/\min(P_B))$ for some constant $\beta$. Plugging it in (27) gives,

$$I(C_i; C_{i+k}, C_{i+2k}, \ldots, C_{i+(M+1)k}) \leq \beta(M+1)\alpha\rho^k \to 0 \\ \text{as } k \to \infty.$$

Combining these inequalities results in

$$I(X_i; X_{i+k}, X_{i+2k}, \ldots, X_{i+(M+1)k}) \\ \leq I(C_i; C_{i+k}, C_{i+2k}, \ldots, C_{i+(M+1)k}) \to 0.$$

Therefore, the MI between $X_i$ and any finite block of sufficiently far future symbols vanishes with a geometric rate in $k$. $\square$

## D. Qualitative Generation Trajectories

This section qualitatively visualizes the generation process of LLaDA-8B on several benchmarks under two decoding strategies: our Dilated Unmasking Scheduler (DUS) and a fixed block scheduler. For each dataset, a representative example is selected, and the full generated answer for four decoding configurations is presented (DUS with two block sizes and fixed scheduling with self-confidence with two block sizes). Tokens are colored according to the diffusion step at which they are first unmasked: early steps appear in blue, intermediate steps in green, and late steps in yellow to red. All prompts are given without their few-shots. The prompt length is reported in tokens for each example.

The background color of each token indicates when it was unmasked during generation:

Early (Stage 1) → Quarter → Middle → Three-Quarter → Late (Final Stage)

## D.1. MATH500 - LLaDA-8B-Instruct

QUESTION (NO FEW-SHOTS)

PROMPT LENGTH: 504 TOKENS

> Problem: A regular pentagon is rotated counterclockwise about its center. What is the minimum number of degrees it must be rotated until it coincides with its original position?

**DUS B=32** (Steps: 11, Generation length: 81)

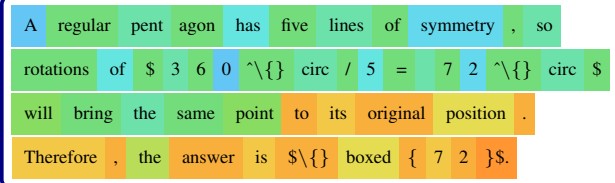

**DUS B=64** (Steps: 7, Generation length: 97)

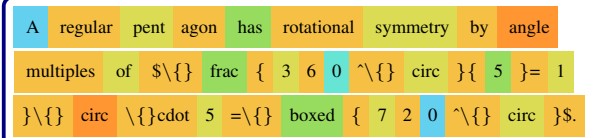

**Fixed B=32** (Steps: 15, Generation length: 72)

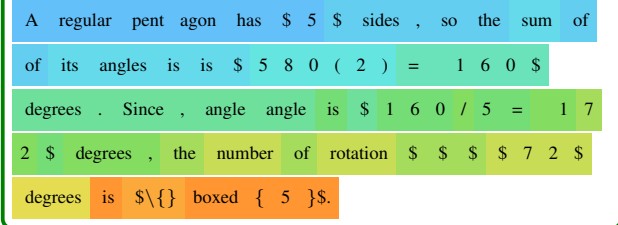

**Fixed B=64** (Steps: 6, Generation length: 25)

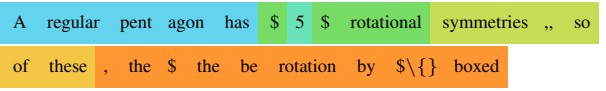

## D.2. GSM8K - LLaDA-8B-Instruct

QUESTION (NO FEW-SHOTS)

PROMPT LENGTH: 1078 TOKENS

> Question: Brandon's iPhone is four times as old as Ben's iPhone. Ben's iPhone is two times older than Suzy's iPhone. If Suzy's iPhone is 1 year old, how old is Brandon's iPhone?
> Answer:

**DUS B=32** (Steps: 15, Generation length: 96)

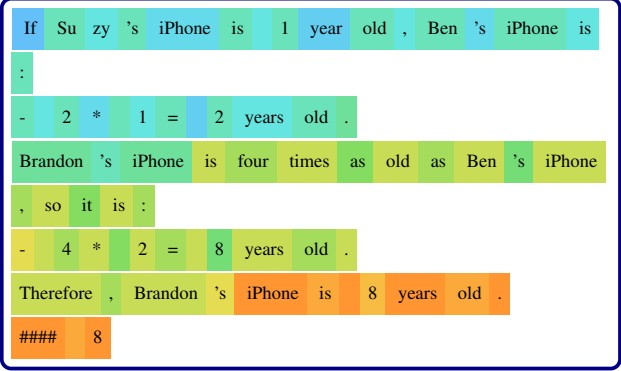

**DUS B=64** (Steps: 12, Generation length: 128)

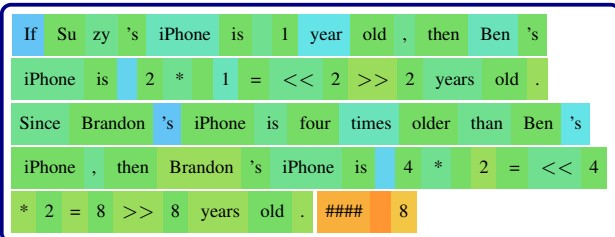

**Fixed B=32** (Steps: 35, Generation length: 198)

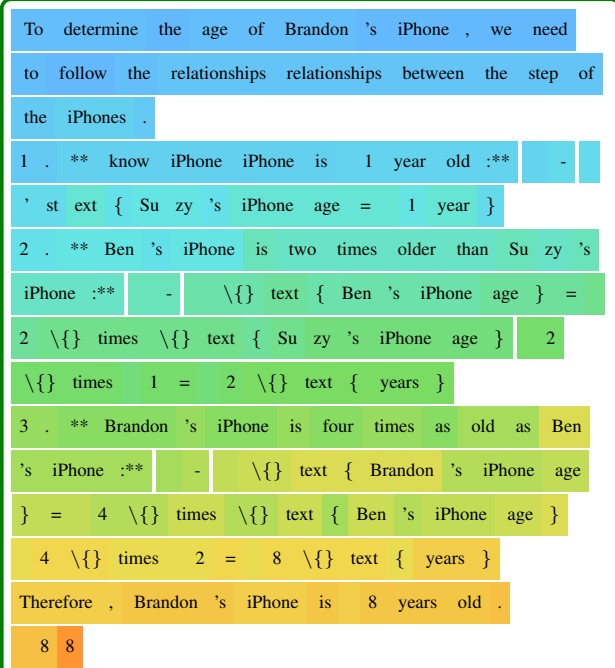

**Fixed B=64** (Steps: 24, Generation length: 253)

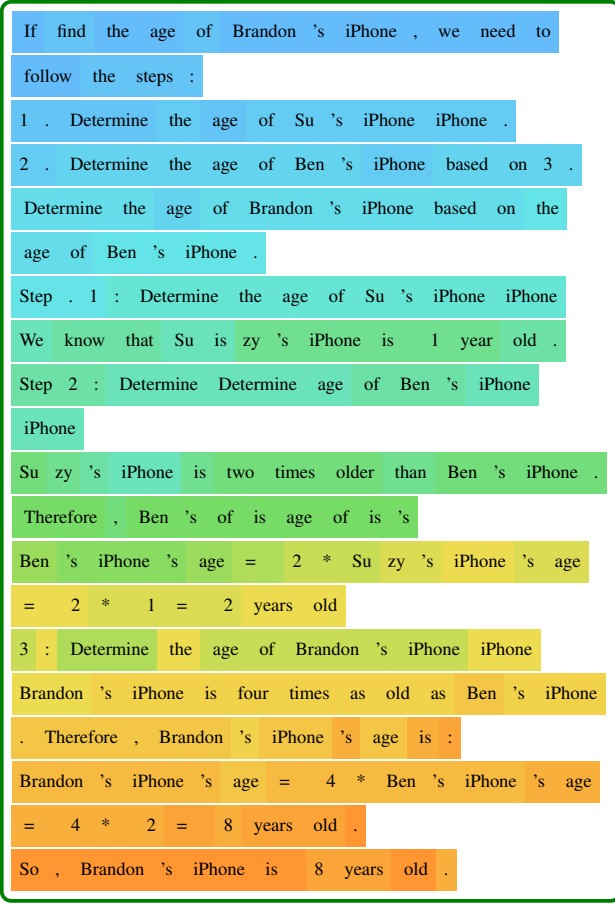

### D.3. HumanEval - LLaDA-8B-Instruct

QUESTION (NO FEW-SHOTS)

PROMPT LENGTH: 71 TOKENS

```
    def string_sequence(n: int) -> str:
    """ Return a string containing space-delimited numbers
    starting from 0 upto n inclusive.
    >>> string_sequence(0)
    '0'
    >>> string_sequence(5)
    '0 1 2 3 4 5'
    """
```

**DUS B=32** (Steps: 5, Generation length: 32)

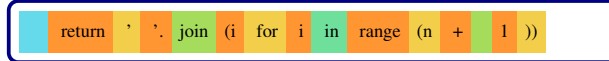

**DUS B=64** (Steps: 7, Generation length: 97)

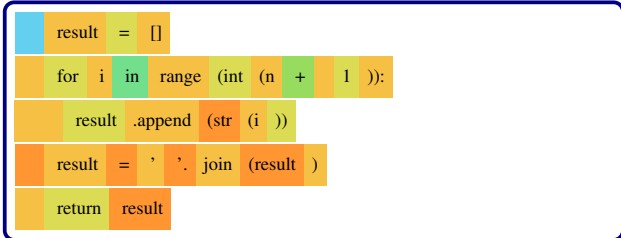

**Fixed B=32** (Steps: 10, Generation length: 44)

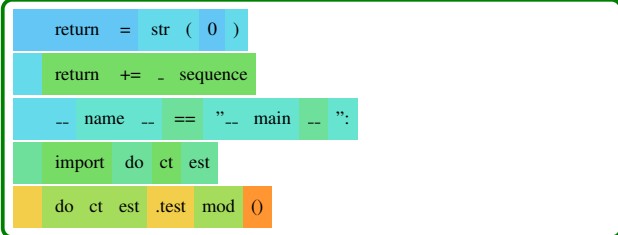

**Fixed B=64** (Steps: 6, Generation length: 19)

## D.4. MBPP - LLaDA-8B-Instruct

QUESTION (NO FEW-SHOTS)

PROMPT LENGTH: 699 TOKENS

> Write a function to find the gcd of the given array elements.
> Your code should pass these tests:
> ```
> assert get_gcd([2, 4, 6, 8, 16]) == 2
> assert get_gcd([1, 2, 3]) == 1
> assert get_gcd([2, 4, 6, 8]) == 2
> [BEGIN]
> ```

**DUS B=32** (Steps: 10, Generation length: 64)

**DUS B=64** (Steps: 7, Generation length: 97)

**Fixed B=32** (Steps: 10, Generation length: 53)

**Fixed B=64** (Steps: 6, Generation length: 30)

### D.5. MMLU-Pro Biology - LLaDA-8B-Instruct

QUESTION (NO FEW-SHOTS)

PROMPT LENGTH: 1400 TOKENS

Question:
Which of the following statements is NOT correct about apoptosis?
Options:
A. Apoptosis, a special type of cell division, requires multiple cell signaling.
B. The fact that apoptosis is widespread across several kingdoms is evidence that it evolved early in the evolution of eukaryotes.
C. Apoptosis plays a crucial role in the development of fingers in embryonic development.
D. Apoptosis prevents an aging and dying cell from damaging neighboring cells.

**DUS B=32** (Steps: 15, Generation length: 96)

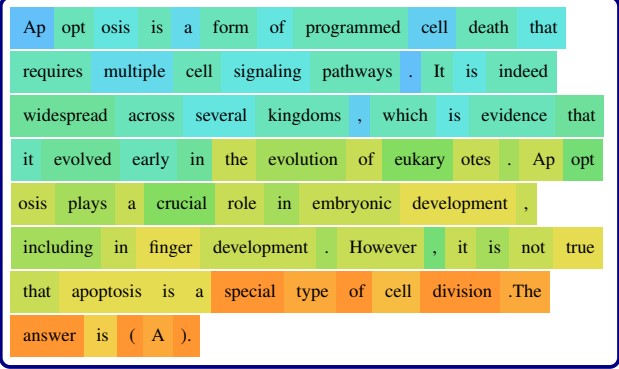

**DUS B=64** (Steps: 13, Generation length: 161)

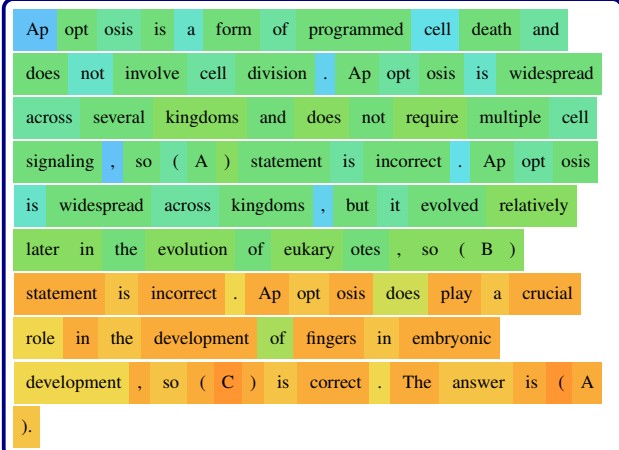

**Fixed B=32** (Steps: 40, Generation length: 256)

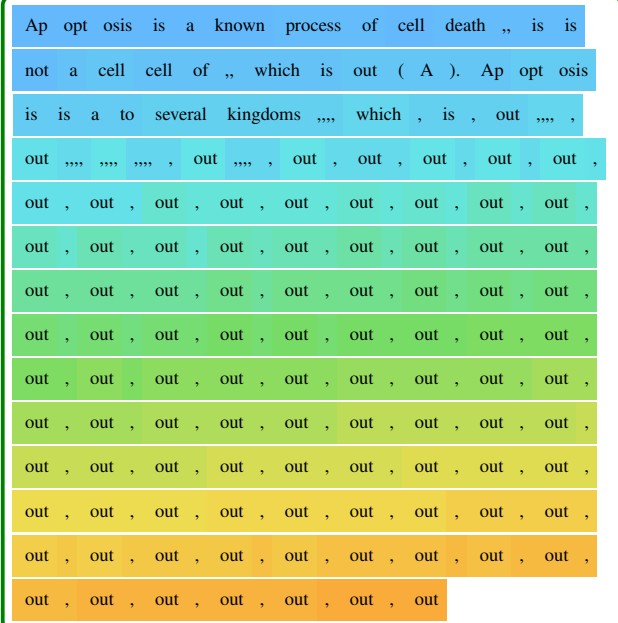

**Fixed B=64** (Steps: 6, Generation length: 44)

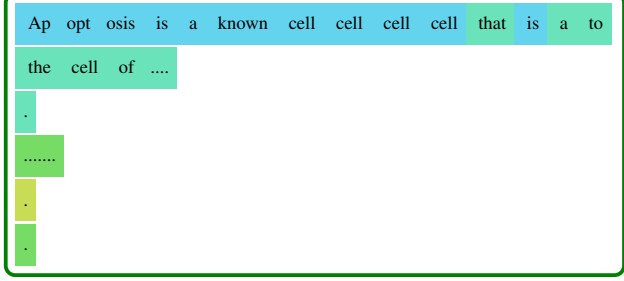

