# OpenReview forum: "Plan for Speed: Dilated Scheduling for Masked Diffusion Language Models"
_ICML.cc/2026/Conference — ICML 2026 regular_

### Official Review · Reviewer_3TfT · 2026-02-24

**Soundness:** 2
**Presentation:** 3
**Significance:** 2
**Originality:** 3
**Overall Recommendation:** 4
**Confidence:** 3

**Summary:**

This paper proposes Dilated Unmasking Scheduler (DUS), an inference-only scheduling method for masked diffusion language models that decides which token positions to unmask in a deterministic, coarse-to-fine dilated pattern, instead of relying on per-token confidence heuristics. The key idea is to unmask well-spaced tokens in parallel to reduce harmful dependencies within each step, which the authors motivate with an entropy-gap analysis and fast-mixing assumptions, yielding roughly O(log B) denoiser calls per block rather than O(B). Empirically, across math, code, general-knowledge, and instruction-following benchmarks, using models such as LLaDA, Dream, and DiffuCoder, DUS consistently outperforms confidence-based planners at matched speedups while preserving predictable inference efficiency. Overall, the paper’s contribution is a drop-in, model-agnostic decoding scheduler that improves the speed–quality tradeoff of MDLM inference without changing model training or architecture.

**Compliance With Llm Reviewing Policy:**

Affirmed.

**Final Justification:**

Thank you for the author's detailed reply, which has resolved my issue. Since my score is already at the acceptance threshold and I don't have a very deep understanding of this research area, I'm keeping my score and confidence.

**Key Questions For Authors:**

1. Can you clarify the practical validity range of the theoretical assumptions (fast-mixing VLMC + near-optimal denoiser) for the real MDLMs used in your experiments, especially on code and long-chain reasoning tasks?
2. How sensitive are the gains to the exact DUS design choices, e.g., base a, block size B, last-set merge rule, and optional skip heuristic, and do you have a recommended tuning rule for practitioners?
3. Can you provide additional evidence that NFE improvements translate into real wall-clock latency / throughput gains in realistic inference settings (batching, hardware kernels, memory constraints)?

**Limitations:**

yes

**Strengths And Weaknesses:**

Strengths:
1. The paper has a clear methodological decomposition (denoiser + planner) and provides a principled entropy-gap framing that explains why naive parallel unmasking can fail, which makes the DUS design rationale technically coherent.
2. The paper is generally well organized, and the figures/tables appear effective at communicating the speed–accuracy tradeoff and qualitative differences in decoding behavior. The notation and scheduler definition are reasonably explicit, including a concrete toy example for DUS.

Weaknesses:
1. The paper is careful and consistent in using NFE as the main efficiency proxy, and DUS clearly improves that proxy. However, real deployment latency and throughput also depend on kernel launch overhead, parallelism utilization, memory bandwidth, KV-cache interaction, and batching behavior. Without wall-clock measurements, the practical magnitude of speed improvements remains somewhat uncertain.
2. The paper itself acknowledges that long-range constraints can increase the entropy gap for any parallel unmasking scheme. This suggests DUS may have regime-dependent effectiveness, but the paper does not yet provide a clear operational guideline for practitioners.
3. Since uncertainty-aware skipping could materially affect both quality and speed, readers may want a clearer understanding of whether DUS’s gains are robust without such heuristics and how sensitive the results are to these choices.

---

> ### Author Rebuttal · Authors · 2026-03-30
>
> We thank the reviewer for the constructive feedback and for recognizing DUS's clear methodological decomposition and effective communication. Due to character limits, we refer to the revised paper for full tables and to our response to Reviewer dBBB for shared experiments (dilated EB/CB tables, wall-clock).
>
> **[W1]** We thank the reviewer for raising this concern. We measured actual end-to-end wall-clock time on RTX 6000 Ada (105 samples, GSM8K, LLaDA-Base):
>
> | Method | Median (ms) | NFE | Speedup | GSM8K |
> |--------|-------------|-----|---------|-------|
> | Llama-3-8B AR | 35,144 | 257 | 0.69x | 49.81% |
> | Token-by-token | 24,280 | 124 | 1x | 72.63% |
> | SC B=16 | 5,783 | 32.9 | 4.2x | 51.23% |
> | DUS B=16 | 6,314 | 32.8 | 3.9x | 59.51% |
> | DUS B=32 | 4,182 | 22.8 | 5.8x | 49.36% |
> | EB gamma=1.0 | 8,577 | 43.3 | 2.8x | 71.72% |
> | CB tau=0.7 | 7,088 | 38.3 | 3.4x | 72.02% |
>
> *NFE measured on 105 samples. Full-dataset accuracy from Tables 1-2.*
>
> Speedups closely match NFE ratios, confirming NFE is a reliable proxy. Model forward = 247ms/NFE (>99.5% cost). Planner overhead is negligible (DUS: 0.0003ms, EB: 1.39ms, CB: 1.88ms). DUS B=16 is 5.6x faster than Llama-3-8B AR.
>
> **[W2]** We agree and added a "Practical considerations" paragraph to Section 4. **Recommended defaults**: base=2, B=16-32 (4-6x NFE speedup). B=16 favors accuracy, B=32 favors speed. DUS outperforms confidence-based planners at any fixed B across all tested domains (Tables 1-2). Our sensitivity analysis (KQ2) confirms these defaults are robust across models and tasks.
>
> **[W3]** We thank the reviewer for highlighting this. All main results (Tables 1-2) use th=0.0 (no skip). DUS is valuable on its own. Skip is an optional enhancement that defers low-confidence tokens to the next block:
>
> LLaDA-Base (MATH500 B=16, HumanEval B=32):
>
> | th | MATH500 (%) | NFE | HumanEval (%) | NFE |
> |----|------------|-----|--------------|-----|
> | 0.0 | 16.8 | 36.4 | 9.76 | 42.8 |
> | 0.3 | 17.2 | 38.8 | **12.80** | 31.3 |
> | 0.5 | **17.6** | 42.2 | 11.59 | 36.8 |
>
> LLaDA-Instruct:
>
> | th | MATH500 (%) | NFE | HumanEval (%) | NFE |
> |----|------------|-----|--------------|-----|
> | 0.0 | **21.0** | 38.3 | 13.41 | 25.6 |
> | 0.5 | 20.6 | 46.4 | **15.24** | 33.9 |
>
> Skip improves code (HumanEval Base: 9.76% to 12.80% at th=0.3) while maintaining reasoning performance. We also integrated DUS-style dilated spacing into EB/CB (see Reviewer dBBB, W2 for full tables). On HumanEval, EB-dilated gamma=4.0 ds=4: +5.5% accuracy at lower NFE than standard EB.
>
> **[KQ1]** We agree the VLMC assumption does not hold literally. A VLMC assumes *variable-length* memory (context l <= L), not short memory. This is what CTW [1] exploits, and Begleiter et al. [2] showed CTW is among the best English text predictors. DUS only needs tokens at distance d to be *on average* less dependent. Our distance analysis shows DUS maintains 2-3x wider spacing with ~100% isolated tokens vs 42-51% for SC on code and reasoning, while achieving equal or higher accuracy. **Near-optimal denoiser**: standard analytical device. Breaks down at high masking rates, which is why DUS uses gradual unmasking.
>
> **[KQ2]** We address each design choice (full tables in revised paper):
> - **base**: base=2 optimal. base=3,4 degrade significantly (3-5x fewer NFE but much worse accuracy).
> - **base_skip**: base_skip=1 (default) consistently better.
> - **block_size**: Graceful tradeoff from B=8 (2.7x) to B=256 single-block (28x). B=16-32 recommended.
> - **merge_last**: Always on. Implementation detail.
> - **skip**: See W3. Optional, most beneficial on code.
>
> **Defaults**: base=2, B=16-32, base_skip=1, threshold=0.0. Robust across all tested tasks and models.
>
> **[KQ3]** See W1. Wall-clock confirms NFE translates directly to latency. DUS's deterministic schedule is amenable to batched inference (all samples follow same pattern). KV-caching (FreeCache, Fast-dLLM, dKV-Cache) is complementary.
>
> [1] Willems et al., CTW Method, IEEE Trans. IT, 1995.
> [2] Begleiter et al., On Prediction Using Variable Order Markov Models, JAIR, 2004.

---

> > ### Author Rebuttal · Reviewer_3TfT · 2026-04-05
> >
> > Thank you for the author's detailed reply, which has resolved my issue. Since my score is already at the acceptance threshold and I don't have a very deep understanding of this research area, I'm keeping my score and confidence.

---

### Official Review · Reviewer_GJTs · 2026-03-12

**Soundness:** 3
**Presentation:** 3
**Significance:** 3
**Originality:** 3
**Overall Recommendation:** 4
**Confidence:** 1

**Summary:**

This paper introduces Dilated Unmasking Scheduling (DUS), a deterministic coarse-to-fine planner for block-wise masked diffusion language models. The key idea is to predefine which positions to unmask at each iteration based on a dilated pattern, reducing the number of denoiser evaluations per block from O(B) to O(log B). The authors provide theoretical analysis showing that spacing indices reduces mutual information between simultaneously unmasked tokens under a fast-mixing VLMC assumption, making parallel unmasking more accurate.

**Compliance With Llm Reviewing Policy:**

Affirmed.

**Key Questions For Authors:**

1 The paper compares DUS against a self-confidence planner with fixed $k = \log{B}$. How does DUS compare against other adaptive schedulers like Jump Your Steps or entropy-bounded sampling? Including such comparisons would help position DUS within the broader literature on diffusion acceleration.

**Limitations:**

The authors acknowledge theoretical limitations but do not discuss how DUS compares to other adaptive schedulers or explore the skip heuristic. A more thorough discussion would strengthen the paper.

**Strengths And Weaknesses:**

## Strength
1. The paper presents a simple yet effective inference-time optimization for masked diffusion models. The key insight—that spacing unmasked positions reduces dependence and enables more accurate parallel prediction—is intuitively appealing and well-motivated. The theoretical analysis, while based on idealized VLMC assumptions, provides a principled justification for why dilated patterns should work.
2. The empirical evaluation is thorough, spanning multiple model families (LLaDA, Dream, DiffuCoder) and diverse task types (math, code, reasoning), with consistent improvements across settings. The speed-quality trade-off analysis (Figure 3) clearly demonstrates DUS's advantages over self-confidence baselines. The method is training-free and easy to implement, making it practically valuable for deployment.
## Weakness
1. The paper compares only against a self-confidence planner, omitting other adaptive schedulers mentioned in related work.
2. Despite the speedups achieved by DUS, Tables 1 and 2 show that even the best diffusion model results often lag behind autoregressive baselines. What do you see as the fundamental limitations of diffusion language models compared to AR models—beyond inference speed—in terms of generation quality, adaptability to instruction tuning, or handling of long-range dependencies? Do you expect these gaps to close with further advances in diffusion architectures, or are there inherent trade-offs?

---

> ### Author Rebuttal · Authors · 2026-03-30
>
> We thank the reviewer for the constructive feedback and for recognizing DUS as a simple yet effective optimization with thorough empirical evaluation. Due to character limits, we refer to the revised paper for full tables and to our response to Reviewer dBBB for shared experiments.
>
> **[W1]** We thank the reviewer for this point. We compared DUS against EB-Sampler (entropy-bounded) and CB-Sampler (confidence-bounded) across all benchmarks on 5 models (LLaDA-Base/Instruct, Dream-Instruct, DiffuCoder-Base/Instruct). Full comparison table in revised paper appendix. The results show a task-dependent landscape: adaptive planners achieve higher accuracy on some domains (particularly code), but their NFE varies per sample and requires per-task calibration of gamma/tau. DUS provides a fixed, predictable schedule (log_a(B) steps per block), enabling uniform batched inference. EB/CB decide *how many* tokens to unmask, while DUS determines *which* positions. These are orthogonal and naturally complementary.
>
> **[W2]** We thank the reviewer for this thoughtful question. The current quality gap stems primarily from training maturity: AR models benefit from years of scaling recipes, RLHF, and data curation, whereas diffusion LMs are trained from scratch with less optimization. However, Prabhudesai et al. [1] show diffusion models outperform AR when data is scarce, suggesting the gap is primarily an engineering challenge, not a fundamental architectural limitation. In code generation, Stable-DiffCoder [2] outperforms its AR counterpart (Seed-Coder, same data/architecture) on HumanEval (86.6 vs 84.8), MBPP (85.7 vs 85.2), and BigCodeBench-Hard (31.8 vs 26.4), demonstrating the gap is closing in structured domains. On our benchmarks, LLaDA+DUS B=16 generates in 6.3s vs Llama-3-8B at 35.1s (5.6x speedup).
>
> **[KQ1]** DUS alone ensures deterministic scheduling with guaranteed position spacing. Compared to EB/CB, DUS achieves competitive accuracy at lower and deterministic NFE on reasoning tasks (see W1), though adaptive planners can outperform on code where per-token confidence signals are strong. We also added dilated EB/CB experiments: applying DUS-inspired spacing as a post-filter on adaptive token selections. On HumanEval (LLaDA-Base, B=32), EB-dilated gamma=4.0 ds=4 achieves 25.61% at NFE=30.8 vs standard EB at 20.12%/NFE=32.0 (+5.5% at lower NFE). Full dilated tables in our response to Reviewer dBBB (W2).
>
> Regarding Jump Your Steps: this method targets score-matching diffusion models (e.g., SEDD [3]), which operate at smaller scale, precluding comparison on our benchmarks.
>
> **[Limitations]** We acknowledge DUS's theoretical guarantees rely on the VLMC assumption, which does not hold universally. DUS's deterministic schedule does not adapt to per-sample difficulty. The skip mechanism provides a first step toward model-conditioned adaptivity, deferring low-confidence tokens to the next block:
>
> LLaDA-Base (MATH500 B=16, HumanEval B=32):
>
> | th | MATH500 (%) | NFE | HumanEval (%) | NFE |
> |----|------------|-----|--------------|-----|
> | 0.0 | 16.8 | 36.4 | 9.76 | 42.8 |
> | 0.3 | 17.2 | 38.8 | **12.80** | 31.3 |
> | 0.5 | **17.6** | 42.2 | 11.59 | 36.8 |
>
> LLaDA-Instruct:
>
> | th | MATH500 (%) | NFE | HumanEval (%) | NFE |
> |----|------------|-----|--------------|-----|
> | 0.0 | **21.0** | 38.3 | 13.41 | 25.6 |
> | 0.1 | 21.2 | 38.3 | 14.02 | 25.2 |
> | 0.5 | 20.6 | 46.4 | **15.24** | 33.9 |
>
> Our dilated EB/CB experiments (Reviewer dBBB, W2) further demonstrate that integrating DUS-style spacing with adaptive planners yields concrete improvements.
>
> [1] Prabhudesai et al., Diffusion Beats Autoregressive in Data-Constrained Settings, 2025.
> [2] ByteDance Seed, Stable-DiffCoder, 2026.
> [3] Lou et al., Discrete Diffusion Modeling by Estimating the Ratios of the Data Distribution, ICML 2024.

---

> > ### Author Rebuttal · Reviewer_GJTs · 2026-04-07
> >
> > Thank the authors for their detailed reply, which has resolved my issue. Since my score is already at the acceptance threshold and I don't have a very deep understanding of this research area, I'm keeping my score and confidence.

---

### Official Review · Reviewer_dBBB · 2026-03-12

**Soundness:** 3
**Presentation:** 3
**Significance:** 2
**Originality:** 3
**Overall Recommendation:** 4
**Confidence:** 4

**Summary:**

The authors propose a parallel inference framework, DUS, that selects tokens according to a deterministic dilated schedule while Pareto dominating confidence-based planners. DUS selects well-separated token positions for coarse-to-fine generation, minimizing the entropy gap between parallel samplers and sequential samplers.

**Compliance With Llm Reviewing Policy:**

Affirmed.

**Final Justification:**

My critical concerns have been sufficiently addressed (the authors confirmed that they use semi-autoregressive sampling preserving local decoding, and that DUS still relies on model confidence in accepting/rejecting token candidates)

**Key Questions For Authors:**

- In practice, how many positions apart are jointly sampled tokens using DUS?
- Can the authors clarify the claim that the denoiser is not optimized to reveal all tokens in a single pass because it is trained to reconstruct masked tokens at each reverse step (line 147-151), and its relation to section 3.3?
- Can the authors motivate why decoding in non-local positions should work well empirically, when a perfect planner cannot be assumed (see weakness 1)
- Can the authors compare the throughput of DUS with the CB-Sampler, the EB-sampler, and AR sampling from an AR transformer?

**Limitations:**

Yes

**Strengths And Weaknesses:**

Strengths
- **Sound conceptual motivation.** The formulation in section 3.3 shows the expected log-likelihood gap between parallel and sequential sampling corresponds to the conditional statistical dependence among sampled tokens. Thus, they motivate parallel sampling of tokens with low conditional dependence
- **Convincing and thorough benchmarking across tasks, models, and inference budgets**. The authors clearly motivate the effectiveness of their sampler, with consistent and meaningful improvements in thorough evaluations. It is clear that the proposed sampler will be impactful in the community, as confidence-based planners have been the dominant approach

Weaknesses
- **Decoding tokens in non-local positions should degrade sample quality, despite paper claims.** The authors argue that parallel samplers should decode tokens at non-local positions to minimize the entropy gap with a sequential planner (lines 199-201 right), but this argument relies on the assumption of a perfect planner. In practice, predicting tokens far from observed context degrades sample quality [4], since predictions have less reliable conditioning. This has been observed in block diffusion decoding [3], where decoding in left-to-right blocks yields a tighter NELBO and better sample quality, suggesting that stronger local conditioning may outweigh the benefits of reduced dependence between jointly sampled tokens
- **Lack of adaptivity in the planner.** The authors claim that DUS Pareto dominates confidence-based sampling, despite being independent of the model output. However, a planner conditioned on the model's predictions could potentially better identify tokens with weak conditional dependence
- **Ambiguous motivating claim.** The authors incorrectly claim that the denoiser is not optimized to reveal all tokens in a single pass because it is trained to reconstruct masked tokens at each reverse step (line 147-151); however, estimating the MDLM NELBO [1,2] involves uniform sampling of masking rates, which means that all tokens can be masked and therefore predicted in one shot. This claim is used to motivate section 3.3 introducing core theoretical results, which makes it difficult to interpret the motivation for this section
- **Insufficient formalizations.** Many definitions are largely introduced at a conceptual level without clear formalizations, making it difficult to follow the authors' theoretical claims
  - The masked diffusion framework is largely introduced at a conceptual level: the denoiser and planner are never explicitly defined. The forward/reverse diffusion processes and their corresponding NELBO are not defined
  - The token sample space is never defined (line 148 left)
  - The expectation in Eq. 3 is not given a precise definition in terms of what distribution it is taken over
- *(minor) presentation concerns:*
   - The term "speedup" is slightly ambiguous in the current definition (despite the explicit definition in Table 4), because it actually corresponds to a theoretical speedup rather than empirical speedup. It would be beneficial if this was rephrased or made more explicit in the tables and figures throughout the paper
  - I noticed there is at least 1 value where DUS underperforms relative to the confidence based sampler, yet DUS is incorrectly bolded (HumanEval w/ Diffucoder-I); the authors should fix this instances of this in the final version
  - some grammatical issues (line 361 left, 357 right)

I am happy to raise my score to 4 if my concerns are addressed, importantly weaknesses #1 and #2 which relate to the core paper motivations

[1] Sahoo et al. Simple and effective masked diffusion language models. NeurIPS 2024

[2] Shi et al. Simplified and generalized masked diffusion for discrete data. NeurIPS 2024

[3] Arriola et al. Block diffusion: interpolating between autoregressive and diffusion language models. ICLR 2025

[4] Kim et al. Train for the Worst, Plan for the Best: Understanding Token Ordering in Masked Diffusions. ICML 2025

---

> ### Author Rebuttal · Authors · 2026-03-30
>
> We thank the reviewer for the constructive feedback. Due to character limits, we refer to the revised paper for full tables.
>
> **[W1]** We agree that proximity to observed context improves prediction quality [1],[2]. Like Block Diffusion [1], DUS operates in semi-AR blocks where every token conditions on the full prefix. Tables 1-2 confirm a clear accuracy-speed tradeoff with larger B. DUS does not eliminate this tradeoff but optimizes *within-block parallelism*: adjacent tokens have the highest mutual dependence, so parallel decoding of clustered tokens introduces the largest approximation error. DUS avoids this and is the safer parallel strategy. Our distance analysis (revised paper appendix) shows DUS achieves 2-3x wider spacing with ~100% isolated tokens vs 42-51% for SC, while outperforming SC at lower NFE.
>
> **Average pairwise distance between co-decoded tokens (LLaDA-Base, 150 samples):**
>
> | Dataset | B | DUS | SC | EB | CB | DUS/SC |
> |---------|---|-----|----|----|------|--------|
> | MATH500 | 16 | **6.22** | 3.00 | 3.49 | 4.47 | **2.08x** |
> | MATH500 | 32 | **9.62** | 4.13 | 4.74 | 7.13 | **2.33x** |
> | HumanEval | 16 | **6.22** | 2.81 | 4.37 | 4.71 | **2.21x** |
> | HumanEval | 32 | **11.59** | 3.71 | 6.39 | 8.63 | **3.12x** |
>
> **% isolated tokens (no adjacent co-decoded tokens):**
>
> | Dataset | B | DUS | SC | EB | CB |
> |---------|---|-----|----|----|------|
> | MATH500 | 16 | **100%** | 51.0% | 64.8% | 70.7% |
> | MATH500 | 32 | **99.1%** | 49.8% | 66.0% | 71.1% |
> | HumanEval | 16 | **100%** | 49.0% | 62.0% | 70.1% |
> | HumanEval | 32 | **100%** | 42.1% | 66.2% | 70.0% |
>
> **[W2]** We agree that model-conditioned planning could help. DUS already incorporates this via the skip mechanism, which defers low-confidence tokens to the next block (HumanEval Base: 9.76% to 12.80% at th=0.3, Instruct: 13.41% to 15.24% at th=0.5; see Reviewer GJTs, Limitations for full skip tables). Beyond skip, we integrated DUS-style dilated spacing into EB/CB. Spacing helps most at aggressive operating points, preventing correlated errors (**preliminary results**):
>
> EB + Dilated Spacing (HumanEval, LLaDA-Base, B=32):
>
> | γ | dilated stride | pass@1 (%) | Avg NFE |
> |---|----------------|-----------|---------|
> | 4.0 | off | 20.12 | 32.0 |
> | 4.0 | 4 | 25.61 | **30.8** |
> | 4.0 | 8 | 27.44 | 45.2 |
> | 1.0 | off | 28.05 | 35.6 |
> | 1.0 | 4 | 28.66 | 43.0 |
> | 1.0 | 8 | **29.88** | 56.8 |
>
> CB + Dilated Spacing (HumanEval, LLaDA-Base, B=32):
>
> | τ | dilated stride | pass@1 (%) | Avg NFE |
> |---|----------------|-----------|---------|
> | 0.5 | off | 26.22 | **20.4** |
> | 0.5 | 4 | 28.66 | 28.8 |
> | 0.5 | 8 | 31.10 | 37.3 |
> | 0.7 | off | **35.37** | 26.4 |
> | 0.7 | 4 | 34.76 | 33.5 |
>
> EB-dilated gamma=4.0 ds=4: +5.5% at lower NFE. Full EB/CB and MATH500 results in revised paper.
>
> **[W3]** We agree the original phrasing was incorrect and revised Section 3.3. MDLM training does include all masking rates. However, the structured inference-time pattern (full prompt + contiguous masked block) is a rare realization under training-time random masking, which is why multi-step unmasking improves quality [2]. Moreover, given that multi-step unmasking is standard in all MDLMs, our entropy-gap analysis determines *which* tokens to unmask at each step.
>
> **[W4]** We agree and revised the paper: (1) token space defined over finite vocabulary, (2) denoiser formally specified, (3) planner given formal signature, (4) Eq.3 distribution made explicit, (5) forward/reverse processes and training objective added in Appendix A.1.
>
> **[W5]** (a) Revised "speedup" to "relative NFE speedup". (b) Bolding error in DiffuCoder-1 corrected. (c) Grammar fixed.
>
> **[KQ1]** With DUS (base=2, B=32), first level places 2 tokens 16 apart; levels halve stride to 2. See W1 tables.
>
> **[KQ2]** See W3.
>
> **[KQ3]** DUS enables an explore-then-revise mechanism: early iterations place exploration tokens at well-separated positions, probing what exists far from context. Later iterations fill gaps with denser context, allowing revision. For reasoning and code, multiple valid solutions exist. Unmasking adjacent tokens together locks in a local sequence that cannot be revised. DUS preserves the model's ability to adjust by ensuring neighbors are unmasked in later iterations.
>
> **[KQ4]** Wall-clock on RTX 6000 Ada (105 samples, GSM8K, LLaDA-Base):
>
> | Method | Median (ms) | NFE | Speedup | GSM8K |
> |--------|-------------|-----|---------|-------|
> | Llama-3-8B AR | 35,144 | 257 | 0.69x | 49.81% |
> | Token-by-token | 24,280 | 124 | 1x | 72.63% |
> | SC B=16 | 5,783 | 32.9 | 4.2x | 51.23% |
> | DUS B=16 | 6,314 | 32.8 | 3.9x | 59.51% |
> | DUS B=32 | 4,182 | 22.8 | 5.8x | 49.36% |
> | EB gamma=1.0 | 8,577 | 43.3 | 2.8x | 71.72% |
> | CB tau=0.7 | 7,088 | 38.3 | 3.4x | 72.02% |
>
> Model forward = 247ms/NFE (>99.5% cost). DUS B=16 is 5.6x faster than Llama-3-8B.
>
> [1] Arriola et al., Block Diffusion, ICLR 2025.
> [2] Kim et al., Train for the Worst, Plan for the Best, ICML 2025.

---

> > ### Author Rebuttal · Reviewer_dBBB · 2026-04-02
> >
> > My concerns have been fully addressed, my main concerns being W1 and W2. I have raised my score to 4.

---

### Official Review · Reviewer_R9sf · 2026-03-12

**Soundness:** 4
**Presentation:** 3
**Significance:** 3
**Originality:** 3
**Overall Recommendation:** 5
**Confidence:** 4

**Summary:**

The authors propose Dilated Unmasking Scheduling, a token selection planner/scheduler for masked diffusion language models.

One exciting capability of text diffusion models is their ability to decode multiple tokens in parallel. However, when tokens are sampled in parallel from an MDLM, they are sampled independently from the product of their conditionals, which may diverge from the model’s true joint conditional distribution. At inference-time, DUS attempts to reduce this error by infilling tokens over a logarithmic number of steps, and following a fixed, widely spaced, dilated pattern.

The authors include a theoretical justification for this algorithm, which assumes that the sequences are sampled from a variable length Markov Chain with fast mixing. In this setting,
mutual information between distant tokens decreases exponentially with distance. This motivates DUS, which avoids selecting nearby indices in the same iteration. Additionally, the entropy of masked positions is expected to decrease later in inference, justifying sampling more tokens in parallel.

The authors evaluate their method on multiple masked diffusion models. They consider math and coding, general knowledge, and instruction following. On these tasks, DUS generally outperforms a widely-used self-confidence planner at higher acceleration factors.

**Compliance With Llm Reviewing Policy:**

Affirmed.

**Final Justification:**

The authors have addressed my concerns. They have improved the soundness and significance of their work through additional comparisons, and by integrating their algorithm into recent dynamic sampling approaches.

This work presents a meaningful contribution to research on diffusion language models.

**Key Questions For Authors:**

How does the DUS compare against EB Sampler and fast dLLM Confidence-aware parallel decoding on reasoning tasks? Including this result would strengthen the soundness of the work.

How does DUS perform when block size equals generation length? Does DUS require block size << generation length to be performant?

How does the skip DUS approach perform on benchmarks?

**Limitations:**

Yes.

**Strengths And Weaknesses:**

## Strengths

1. The paper introduces a simple and motivated parallel scheduler for sampling from masked diffusion models.

2. DUS is evaluated on multiple benchmarks, where the method improves on self-confidence based parallel decoding. The authors also run key ablation studies.


3. The authors provide a theoretical justification for the method.

4. An exciting capability of masked diffusion models is their ability to decode multiple tokens in parallel. However, this tends to significantly hurt performance. This paper shows that dilated sampling methods offer a potential way to mitigate some of this degradation.

5. The writing of this paper is clear, and easy to follow.


## Weakness

1. For their theoretical analysis, the authors assume that the generative process for the data is a fast-mixing VLMC. This assumption is not realistic for many settings, including coding, poetry, and reasoning tasks. The authors acknowledge this.

2. It is well established that parallel self-confidence based decoding yields poor performance. Algorithms like EB sampler and confidence-aware parallel decoding are significantly stronger alternatives. However, the authors only compare to these stronger baselines on IFEval.


3. Finally the authors propose a “skip” variant that defers high entropy positions, but I could not find an empirical evaluation of the method.

Notes:

In the abstract, the authors note that self-confidence based decoding reduces to autoregressive decoding, but this finding is not directly supported in the paper. Engaging with related prior work that analyzes this behavior, like [1] and/or [2] could strengthen the motivation.

[1] https://arxiv.org/abs/2506.20639
[2] https://arxiv.org/abs/2510.19990

---

> ### Author Rebuttal · Authors · 2026-03-30
>
> We thank the reviewer for the constructive feedback and for recognizing DUS as a simple, motivated parallel scheduler with clear writing. Due to character limits, we refer to the revised paper for full tables and to our response to Reviewer dBBB for shared experiments.
>
> **[W1]** We agree that the fast-mixing VLMC assumption does not hold literally for all domains. A VLMC assumes *variable-length* memory (each position has context length l <= L), not short memory everywhere. Most tokens have short effective context, while some (variable reuse in code, long-range reasoning) require longer context. This variable-depth structure is what compression algorithms such as CTW [1] exploit, and Begleiter et al. [2] showed CTW is among the best predictors for English text across six VLMC-based algorithms. The fast-mixing property (geometric MI decay) is strictly stronger than what DUS requires. DUS only needs tokens spaced at distance d to be *on average* less dependent, not that every pair is independent. Our distance analysis validates this on VLMC-violating tasks: across 4 planners on HumanEval (code) and MATH500 (reasoning), DUS maintains 2-3x wider average spacing with ~100% isolated tokens vs 42-51% for SC, while achieving equal or higher accuracy. Full distance and isolation tables in the revised paper.
>
> **[W2]** We agree EB and CB are stronger baselines than SC. We compared EB/CB across all benchmarks on 5 models (full table in revised paper). The results show a task-dependent landscape: adaptive planners achieve higher accuracy on code, but their NFE varies per sample and requires per-task calibration. DUS provides deterministic scheduling. On GSM8K (LLaDA-Base), DUS B=16 averages 32.6 NFE at 59.51%, while SC B=16 averages 99.0 NFE for 51.23%, illustrating SC's near-sequential collapse [3],[4]. EB/CB adaptively change the number of tokens to unmask, while DUS enforces position spacing. These are orthogonal. We also integrated DUS-style dilated spacing into EB/CB (see our response to Reviewer dBBB, W2 for full tables and analysis). On HumanEval, EB-dilated gamma=4.0 ds=4 achieves 25.61% at NFE=30.8 vs standard EB at 20.12%/NFE=32.0 (+5.5% at lower NFE).
>
> **[W3]** We thank the reviewer for highlighting this. We evaluated the skip mechanism on MATH500 (B=16) and HumanEval (B=32) with both Base and Instruct. All main results (Tables 1-2) use th=0.0 (no skip). DUS is valuable on its own. Skip is an optional enhancement that defers low-confidence tokens to the next block:
>
> LLaDA-Base:
>
> | th | MATH500 (%) | NFE | HumanEval (%) | NFE |
> |----|------------|-----|--------------|-----|
> | 0.0 | 16.8 | 36.4 | 9.76 | 42.8 |
> | 0.3 | 17.2 | 38.8 | **12.80** | 31.3 |
> | 0.5 | **17.6** | 42.2 | 11.59 | 36.8 |
>
> LLaDA-Instruct:
>
> | th | MATH500 (%) | NFE | HumanEval (%) | NFE |
> |----|------------|-----|--------------|-----|
> | 0.0 | **21.0** | 38.3 | 13.41 | 25.6 |
> | 0.5 | 20.6 | 46.4 | **15.24** | 33.9 |
>
> Skip improves code generation (HumanEval Base: 9.76% to 12.80% at th=0.3) while maintaining reasoning performance.
>
> **[KQ1]** See W2. DUS provides deterministic speedup while EB/CB offer higher accuracy at variable NFE.
>
> **[KQ2]** DUS does not require block_size << gen_length. In single-block mode (B=gen_length, 9 NFE, 28x speedup):
>
> | Model | MATH500 | GSM8K | HumanEval |
> |-------|---------|-------|-----------|
> | Base th=0.0 | 8.2% | 12.51% | 9.76% |
> | Base th=0.3 | 7.0% | **17.44%** | **10.98%** |
> | Instruct th=0.0 | **17.2%** | **42.30%** | 14.63% |
> | Instruct th=0.3 | 10.4% | 37.91% | **16.46%** |
>
> Multi-block (B=32) remains best quality-speed tradeoff, but single-block is viable when maximum throughput is needed.
>
> **[KQ3]** See W3. Skip improves code accuracy while maintaining reasoning performance.
>
> **[Notes]** We added citations to Gong et al. [3] and Horvitz et al. [4] in Section 2, supporting that SC planners collapse to near-sequential L-to-R generation. This motivates DUS: a deterministic dilated schedule guarantees true parallel decoding by construction.
>
> [1] Willems et al., CTW Method, IEEE Trans. IT, 1995.
> [2] Begleiter et al., On Prediction Using Variable Order Markov Models, JAIR, 2004.
> [3] Gong et al., Entropy Sink, 2025.
> [4] Horvitz et al., Is Parallel Decoding Actually Parallel?, 2025.

---

> > ### Author Rebuttal · Reviewer_R9sf · 2026-04-03
> >
> > Thank you for providing additional experiments that validate the skip mechanism, and comparing to/integrating dynamic samplers.
> >
> > I am increasing my score accordingly.
> >
> > Small notes:
> > - I suspect that the block size = gen length performance drop on GSM8k is due to DUS unmasking an answer at the end of the sequence before the reasoning trace. This is a setting where the VLMC fast-mixing assumption does not hold.
> > - The titles in citations [3] and [4] are not correct. They are "DiffuCoder: Understanding and Improving Masked Diffusion Models for Code Generation" and "No Compute Left Behind: Rethinking Reasoning and Sampling with Masked Diffusion Models" respectively.

---

> > > ### Author Response · Authors · 2026-04-04
> > >
> > > We thank the reviewer for the positive assessment and for increasing the score.
> > >
> > > **Citation corrections:** We apologize for the incorrect titles in [3] and [4]. The correct titles are: [3] Gong et al., "DiffuCoder: Understanding and Improving Masked Diffusion Models for Code Generation," 2025; [4] Horvitz et al., "No Compute Left Behind: Rethinking Reasoning and Sampling with Masked Diffusion Models," 2025. We have corrected these in the revised paper.
> > >
> > > **VLMC and single-block GSM8K:** We agree with the reviewer's hypothesis. We have added a note on this in the revised paper's limitations discussion.

---

### Decision · Program_Chairs · 2026-04-30

**Decision:**

Accept (regular)

**Comment:**

This paper introduces Dilated Unmasking Scheduler (DUS) for masked diffusion language models. DUS picks tokens to unmask in parallel according to a deterministic coarse-to-fine dilated schedule at inference time. The resulting well-spaced parallel token generation can reduce denoiser calls per block to a logarithmic number of steps. DUS is theoretically justified based on the variable-length Markov chain with fast mixing. Extensive experiments conducted on a variety of benchmarks show the effectiveness of the proposed DUS in terms of both efficiency and performance over existing confidence-based planners. Overall, this is an interesting paper. The idea is sufficiently novel. The work is theoretically solid with strong performance, which may have its valuable to the machine learning community. Most of the raised concerns have been cleared by the authors' rebuttal. All reviewers are supportive of accepting it. The authors need to revise the paper according to the discussion with reviewers in its final version.